# Peritectic titanium alloys for 3D printing

Pere Barriobero-Vila[1], Joachim Gussone[1], Andreas Stark [2], Norbert Schell[2], Jan Haubrich[1] & Guillermo Requena[1,3]

Metal-based additive manufacturing (AM) permits layer-by-layer fabrication of near net-shaped metallic components with complex geometries not achievable using the design constraints of traditional manufacturing. Production savings of titanium-based components by AM are estimated up to 50% owing to the current exorbitant loss of material during machining. Nowadays, most of the titanium alloys for AM are based on conventional compositions still tailored to conventional manufacturing not considering the directional thermal gradient that provokes epitaxial growth during AM. This results in severely textured microstructures associated with anisotropic structural properties usually remaining upon post-AM processing. The present investigations reveal a promising solidification and cooling path for $\alpha$ formation not yet exploited, in which $\alpha$ does not inherit the usual crystallographic orientation relationship with the parent $\beta$ phase. The associated decrease in anisotropy, accompanied by the formation of equiaxed microstructures represents a step forward toward a next generation of titanium alloys for AM.

[1] Institute of Materials Research, German Aerospace Center (DLR), Linder Höhe, 51147 Cologne, Germany. [2] Helmholtz-Zentrum Geesthacht, Max-Planck-Straße 1, 21502 Geesthacht, Germany. [3] Metallic Structures and Materials Systems for Aerospace Engineering, RWTH Aachen University, 52062 Aachen, Germany. Correspondence and requests for materials should be addressed to P.B.-V. (email: pere.barrioberovila@dlr.de)

Metal-based additive manufacturing (AM)—colloquially termed metal 3D printing—is resulting in a paradigm change across multiple industries, such as the aerospace, biomedical, and automotive sectors. One of its key strengths is the fabrication of near net-shape metallic components with complex geometries providing, e.g., inner channels for cooling fluids, or bionic and load-optimized structures of minimal weight not achievable with conventional production methods like casting or machining. Layer-by-layer AM production from a 3D computer-aided design provides design freedom, increased product customization, and shorter time to market[1,2]. For titanium-based components, these advantages account for estimated production savings up to 50%, by basically missing out exorbitant machining costs and material loss[3]. In aerospace, this focuses on parts with high buy-to-fly ratio (BTF): the weight of the purchased stock material to that of the finished part. Typical aerospace components can have the BTF of 10:1, 20:1, and even 40:1 using conventional manufacturing processes. AM is capable to reduce it close to 1:1. For instance, 50% reduction of production costs has been reported for a wrought Ti-6Al-4V engine bracket using AM[4]. AM also allows the repair of expensive titanium-based components (e.g., flanges, fan blades, vanes, and landing gears) at 20–40% of new parts cost[1]. AM weight-optimized components can imply a progress of environmental targets. Previous studies concluded on saving 3.3 million litres of fuel over the aircraft's life, obtained by a 55% weight reduction using AM Ti-6Al-4V seat buckles[1].

A critical issue for acceptance and certification of AM parts is the degree of isotropy of their microstructure derived from the solidification conditions during AM and eventual post-treatments[1]. A deep-rooted drawback during AM of Ti-alloys is the steep, directional thermal gradient in the molten metal pool, which prevents nucleation ahead of the solidification front, provoking epitaxial growth across solidified layers[1,5]. This is particularly relevant for powder-bed AM techniques, such as selective laser melting (SLM). The typical resulting microstructures are coarse, columnar prior β grains with strong <100 > β orientation along the building direction, normal to synthesized powder layers[5,6]. This effect is well known to occur in the popular α + β Ti-6Al-4V alloy, which accounts for more than 50% of the titanium market[7] and leads—by far—AM of Ti alloys[8].

Owing to the complicated thermal history undergone by materials during SLM, namely sharp cycles of steep heating ($\sim 10^6$–$10^7$°C s$^{-1}$) and cooling (>$10^3$°C s$^{-1}$) rates[9], brittle martensitic microstructures unsuitable for structural applications are usually obtained via diffusionless transformation of parent β grains (primary high temperature phase) in the as-built condition of α + β Ti alloys. For instance, α′ martensite formation in Ti-6Al-4V occurs for cooling rates above ~410 °C s$^{-1}$[10]. Though in Ti alloys both α′ and the stable α phase present a hexagonal close-packed (hcp) lattice, the low ductility ( < 10%) and fracture toughness exhibited by martensitic microstructures upon AM manufacturing is mainly a consequence of high density of defects (e.g., dislocations, twins) present in the α′ phase[11,12]. Differently, the brittleness resulting from martensite formation in steels is associated with the distorted body-centered cubic tetragonal lattice containing ordered arrangements of interstitial C atoms[13].

Post-thermal and/or thermomechanical treatments are commonly applied to the as-built AM components to improve the strength–ductility trade-off. This can include supertransus or subtransus heat treatments[14,15], as well as hot isostatic pressing[16] inducing formation of stable α and β via decomposition of metastable microstructures. Subtransus treatments have limited impact on the microstructure and columnar morphologies derived from epitaxial growth are usually maintained. During supertransus treatments, rapid growth of β takes place, leading to

excessive grain growth and coarsening[15]. Apart from representing a costly methodology that reduces the economical attractiveness of AM, these post-treatments do not represent an alternative to mitigate crystallographic texture and its effect on mechanical performance of the alloys[17–19].

Approaches to tackle epitaxial growth in AM Ti alloys include B addition to α + β and β compositions. For instance, the effect of B on powder blends of Ti-20V, Ti-12Mo[20], and Ti-6Al-4V powder alloy[21,22] can result in grain refinement. Microstructure globularization preserving the Burgers orientation relationship (OR) between α and β phases has been reported[20]. The use of B leads to formation of ceramic TiB needles. Thus, the presence of TiB has been associated with strength increase at expenses of ductility, as well as localized plastic flow and damage caused by inhomogeneous distributions of the TiB needles[21,23]. Other investigations with a SLM-produced Ti-1Al-8V-5Fe alloy showed globularization and formation of small β grains along the building direction, owing to the presence of Fe[24]. In addition to inhomogeneous distribution of α laths resulting from partitioning and segregation of Fe, the as-built microstructure obtained was formed by α films along β grain boundaries, which are detrimental to fatigue and ductility in β Ti-alloys[25].

A further strategy addressing reduction of anisotropy and improvement of the strength–ductility trade-off during metal-based AM consists in exploring the possibilities of the intrinsic heat treatment (IHT), namely the thermal history induced by the heating source (e.g., laser) to previously deposited layers. As shown for Ti-6Al-4V, intensified IHT can generate extensive martensite decomposition leading to configurations of stable α and β phases in a single AM process step[6,26]. For precipitation-hardened alloys, the IHT can provoke the formation of finely dispersed second phase particles and therefore, offers the possibility to shorten or avoid subsequent aging treatments[27]. Illustrative examples are homogeneous dispersions of fine $N_3Ti$ and NiAl precipitates obtained in maraging steels, as well as of $Al_3(Sc, Zr)$ particles in Al–Sc alloys[27,28]. Besides dispersion-hardening effects, the incorporation of nanoparticles permitted to avoid large columnar grains and cracking during SLM of high-strength Al alloys. These nucleants promoted fine-grain formation, resulting in strengths comparable to wrought materials[2].

On the other hand, AM opens up the opportunity to control liquid–solid, as well as solid–solid phase transformations owing to the highly localized solidification and thermal cycles applied by a heat source in motion[18,19]. With this purpose, the present work aims at tailoring Ti alloys to the metallurgical conditions of AM by exploiting metastability around peritectic and peritectoid reactions. The findings obtained reveal a promising path of α formation not exploited yet to decrease the texture in as-built, as well as post-treated AM Ti alloys.

## Results

**Ti–La system**. Our approach consists in adding the solute α-stabilizer La to commercially pure titanium (CP Ti) aiming at altering the regular Burgers-related β → α transformation. According to the current knowledge of the Ti–La equilibrium phase diagram (shown partially in Fig. 1a and Supplementary Fig. 1), at 2 wt.% La the Ti–La system presents during cooling two paths of α formation after passing through a $L_1 + β →$ La-bcc peritectic reaction[29]: La-bcc + β → α(peritectoid) and La-bcc → La-fcc + α (eutectoid). The response of the Ti-2wt.% La (Ti-2La) alloy is studied here under AM conditions reproduced by the powder bed-based technique SLM described in reference[8].

Upon manufacturing Ti-2La and CP Ti using identical SLM strategies (see Methods), corresponding microstructures with twisted α grains describing tortuous trajectories (Fig. 1b) and

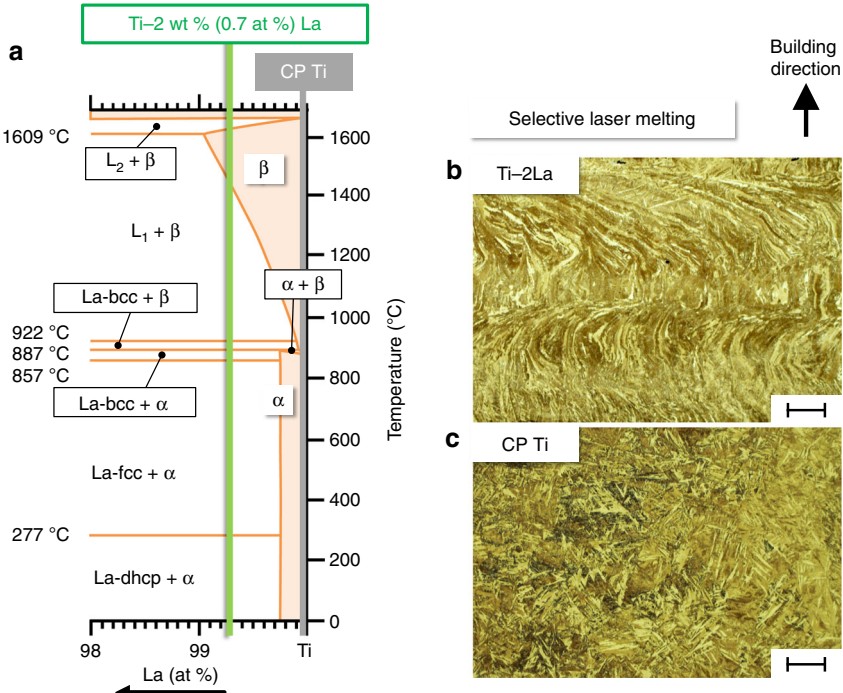

**Fig. 1** Additive manufacturing of the Ti–La system. The approach of selecting a composition Ti-2wt.% La permits to explore an uncommon path of α formation in titanium alloys, by altering the regular Burgers-related β → α transformation. **a** Portion of the Ti–La phase diagram adapted from[29,50], indicating the compositions used for selective laser melting. **b**, **c** Overview of the as-built microstructures for these compositions, namely commercially pure titanium grade 1 (CP Ti) taken as reference, and the Ti-2wt.% La alloy (Ti-2La), respectively. Scale bars, 100 μm

homogeneously distributed α′ martensite plates (Fig. 1c) are obtained. The microhardness of the highly martensitic CP Ti is $229 \pm 2.5$ HV$_{0.2}$, while that of Ti-2La reaches $251 \pm 2.4$ HV$_{0.2}$, although the latter microstructure mainly comprises stable α phase. Bulk texture measurements of these conditions using high-energy synchrotron X-ray diffraction (HEXRD) (Fig. 2a, b) reveal a higher preferential orientation of the hcp lattice of CP Ti, with the c-axis slightly tilted between 42.4–49° with respect to the SLM building direction. These values are close to 45° given by the Burgers OR {002}α ∥ {110}β when preferential <100> β orientation occurs along the building direction[30]. The texture results are indicated as α-Ti for simplicity, since α′ and α have hcp crystal structures. This type of texture satisfying the complementary Burgers OR with <100> β orientations is a consequence of the texture inherited from the parent β phase via martensitic transformation, as can be observed in the magnified region at the bottom of Fig. 2c. There, reconstruction of parent β grains (see Methods) shows typical α′ plates extending toward the interior of parent β grains from their boundaries, according to the Burgers OR between the phases[30]. On the other hand, the less pronounced texture in as-built Ti-2La can be explained by the hierarchical microstructure obtained for this alloy, containing extensive distributions of large tortuous and fine-equiaxed α grains (Fig. 2d). Minor agglomerations of α′ plates can also be seen as pointed by arrows in the magnified region of Fig. 2d (see Fig. 2e). Arrangements of fine α grains can be observed between layers of elongated α grains marked between discontinuous lines in Fig. 2d.

Columnar-to-equiaxed transitions (CETs) of grain formation have been related to spatial–temporal variations in the thermal gradient (G) and solidification rate (R), as presented in G-R solidification maps[31]. These terms strongly depend on local composition, which is governed by liquid/solid interfacial instabilities (e.g., constitutional undercooling, CU)[18,31]. For the hierarchical microstructure shown in Fig. 2d, CU may be able to provide the necessary driving force for grain nucleation leading to regions with fine equiaxed grains. It seems plausible to suggest that CU variations can take place in heat-affected regions promoting the formation of fine equiaxed α grains: cyclic re-melting of previously deposited material may improve the compositional homogeneity of the Ti-2La powder blend and create new nuclei via CU. The low solubility of La in α-Ti and β-Ti may as well contribute to a large CU[29]. Although columnar grain growth is observed in Fig. 2d, zooming in a representative region of Fig. 2d (see Fig. 2e) reveals that fine equiaxed α grains (<10 μm) coexist with the elongated and tortuous α grains, frequently decorating their boundaries. Thus, nucleation ahead of the solidification front, i.e., local alteration of the solidification mode, may be induced in these regions (Fig. 2e). Furthermore, the alternating microstructure of Fig. 2d is probably also influenced by the checkerboard SLM strategy, since the scanning direction of the laser, i.e., the thermal gradient front of the melt pool, changes for each layer (Methods). Changes in the crystallographic orientation of grains have been reported for AM powder-bed techniques by changing the processing parameters[32]. These effects do not occur for CP Ti.

These microstructural features in Ti-2La are not characteristic of typical transformation mechanisms usually observed in CP Ti or α + β Ti-alloys, namely martensite β → α′ transformation or sympathetic nucleation and growth, leading to parent β grains filled with Widmanstätten structures of α′ plates and/or α lamellae (i.e., basket weave microstructures commonly encountered in α + β Ti alloys)[25,33].

**Phase transformation kinetics.** In order to gain understanding of the transformation mechanisms involved in the Ti-2La alloy, in situ HEXRD was performed to track the phase transformation kinetics during post-thermal treatment of the SLM as-built state (see Methods). Upon cooling with 20 °C min$^{-1}$ from 950 °C (L$_1$

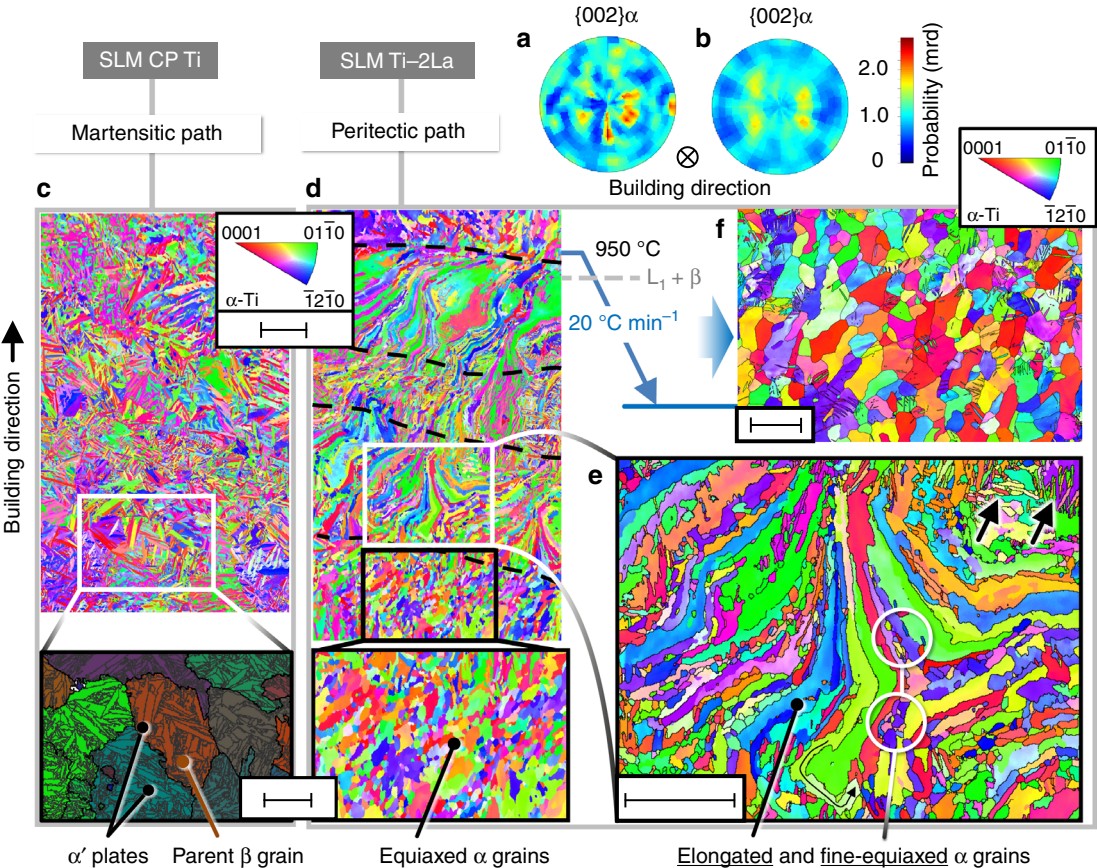

**Fig. 2** Texture modification during selective laser melting (SLM) of Ti-2wt.% La (Ti-2La). The addition of 2wt.% La to commercially pure titanium grade 1 (CP Ti) leads to an attenuation of the preferential orientation of α phase along the building direction as shown in **a**, **b**: normalized pole figures of {002}α reconstructed from a gauge volume of $1 \times 1 \times 5$ mm$^3$ for CP Ti and Ti-2La, respectively. **c** The martensitic microstructure of CP Ti consists of α′ plates extending within parent β grains according to the Burgers OR. Texture attenuation occurs as a consequence of arrangements of small equiaxed α grains with multiple orientations, nucleated in Ti-2La as shown in **d** and **e** (e.g., see encircled grains). **f** Post- thermal treatment of the SLM as-built condition via slow cooling with 20 °C min$^{-1}$ from 950 °C passing through the peritectic line (i.e., from L$_1$ + β field down to room temperature) provokes the formation of new α grains and extensive globularization, leading to a recrystallized-like microstructure. Black lines in **e**, **f** indicate high-angle grain boundaries (misorientation >10°). The scale bars in **c**, **d** and in its magnified regions (at the bottom) are 100 μm and 50 μm, respectively; in **e** and **f**, 50 μm

+ β field) down to room temperature (RT), i.e., under thermal conditions closer to thermodynamic equilibrium than SLM, formation of new α grains and extensive globularization resulting in a recrystallized-like microstructure occurs (Fig. 2f). Moreover, grain refinement takes place by increasing the cooling rate, as clearly inferred from the 5 and 100 °C min$^{-1}$ conditions compared in Fig. 3. The HEXRD results presented in Fig. 4 and the discussion in the following lines point to heterogeneous nucleation of α via the peritectic path L$_1$ + β → α as the main mechanism responsible for the microstructure obtained for the SLM Ti-2La alloy before and after heat treatment.

During the initial stage of cooling between 950–900 °C, slight formation of α takes place in the L$_1$ + β state, as indicated by an increase of 6 vol.% of α visible in the evolution of phase volume fractions (diagram on the right of Fig. 4a). As temperature decreases, a rapid transformation β → α leading to ~95 vol.% of α in ~3.7 min takes place between 900–850 °C. Extensive formation of α reflections increasing their intensity during transformation suddenly occurs at about 900 °C, as shown in Fig. 4b. The presence of L$_1$, β, and α, as well as the absence of La reflections at the beginning of the transformation (905 °C) can be directly identified from the raw diffraction image in Fig. 4c. This points to the formation of α prior to the expected peritectic L$_1$ + β → La-bcc reaction given by the up-to-date knowledge of the equilibrium Ti–La phase diagram (Fig. 1a). The first formed La-bcc reflections

appear at 875 °C from the melt (L$_1$) as shown in the raw diffraction image of Fig. 4c. According to the phase quantification of HEXRD spectra (Fig. 4a), the peritectic reaction L$_1$ + β → La-bcc starts after about 50% completion of the β → α transformation. Metallographic analysis of the SLM Ti-2La alloy quenched from 950 °C (L$_1$ + β field) down to RT (Methods), with a cooling rate of ~85 °C s$^{-1}$ between 950–350 °C, is shown in Fig. 4d. Some La particles are visible in white along seams fragmenting prior β grains, possibly as a product from the La-rich liquid (L$_1$). Moreover, the presence of small α particles pointed by arrows can be seen at former β/L$_1$ interfaces. These particles present a slightly brighter contrast than prior grains. This, together with the results obtained from in situ HEXRD analysis described for Fig. 4a–c suggest that the nuclei of α form at the β/L$_1$ interfaces and expand during subsequent β → α transformation.

The possibility that nanometric La nuclei not detectable by HEXRD may serve as nucleation sites for α (e.g., via peritectoid reaction La-bcc + β → α) is considered hereafter. Supplementary Fig. 2 shows the microstructure obtained upon cooling from the La-bcc + β field at 900 °C with 20 °C min$^{-1}$, i.e., by minimizing the influence of L$_1$ on α formation according to the equilibrium phase diagram (Fig. 1a). Here, tortuous grain arrangements remaining from the as-build state seen in Fig. 1b and Fig. 2d can still be observed, while this effect is not visible for the 5, 20, and 100 °C min$^{-1}$ conditions cooled from 950 °C, where

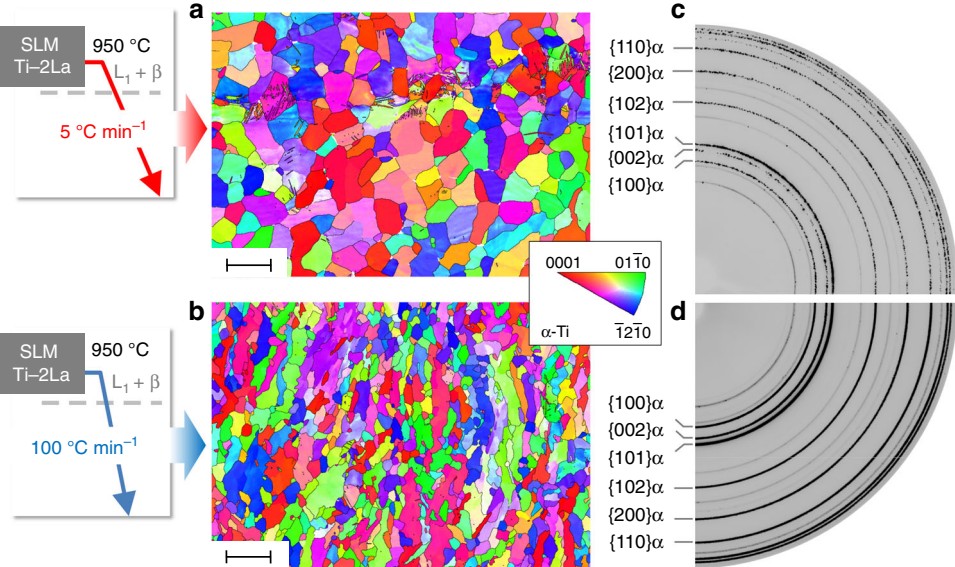

**Fig. 3** Grain refinement in the Ti-2wt.% La (Ti-2La) alloy after selective laser melting (SLM). Post thermal treatment of the SLM as-built condition by cooling with **a** 5 °C min$^{-1}$ and **b** 100 °C min$^{-1}$ from 950 °C (L$_1$ + β field) down to room temperature, results in the formation of new α grains of smaller size with increasing cooling rate and extensive globularization. Scale bars, 50 μm. **c**, **d** Representative quarters of Debye–Scherrer rings obtained for a gauge volume of 1 × 1 × 5 mm$^3$ and the microstructures shown in **a**, **b** respectively. The spotty rings obtained for 5 °C min$^{-1}$ compared with the continuous {hkl}α rings shown for 100 °C min$^{-1}$, reflects for the latter a remarkably smaller grain size in the bulk of the alloy. Black lines in **a**, **b** indicate high angle grain boundaries (misorientation >10°)

recrystallized-like microstructures with new α grains are obtained (Fig. 2f and Fig. 3a, b). This correlates well with the phase transformation kinetics provided by in situ HEXRD, and points to a negligible influence of La particles on α formation compared to the peritectic L$_1$ + β → α path.

The pole figures in Fig. 4e for {002}α and {110}β at 850 °C, i.e., at the end of the β → α transformation, show that α forms to a large extent with orientations distinct from those given by the strong poles shown by β. This indicates that a fraction of α does not inherit the texture of β. This, together with local crystallographic orientation analyses of adjacent α-β grains in the condition obtained upon cooling down to RT (Supplementary Fig. 3), indicates that the usual Burgers OR relationship {002}α || {110}β typical for β → α transformation is partially avoided.

The equilibrium invariant reactions of Ti-2La according to the equilibrium phase diagram are contrasted in Table 1 with the transformations identified by in situ HEXRD in the present study (Fig. 4). The peritectic path of α formation observed here for Ti-2La (L$_1$ + β → α) is analogous to that occurring in γ-TiAl alloys, where interdendritic regions solidify under non-equilibrium conditions via L + β → α[34–36]. During this reaction, peritectic α grows at solid/liquid interfaces driven by super-saturation of the liquid[34,35]. This mechanism seems to correspond to that deduced from Fig. 4d. Moreover, for Ti-2La, the remaining liquid undergoes the peritectic L$_1$ + β → La-bcc reaction. Previous investigations of γ-TiAl alloys suggested that the peritectic α nucleates from the melt and not at the parent β matrix[37,38]. Accordingly, formation of α from liquid via L → L + α transformation was shown to take place during fast solidification (1.25·10$^3$ °C s$^{-1}$) of a Ti-48 at.% Al alloy[39]. This implies that the formation of new α orientations is related to its own preferential growth direction and not to that of the parent β phase. For Ti-2La, this effect is corroborated in the EBSD analysis presented in Supplementary Fig. 3, in the pole figures of Fig. 4e after β → α transformation, and is reflected in the formation of many diverse α orientations present in the as-built (Fig. 2d, e) and

post-treated Ti-2La alloy, leading to highly equiaxed microstructures (Fig. 2f and Fig. 3a, b). Formation of α from the melt, i.e., ahead of the SLM solidification front, might explain the presence of tortuous primary α grains observed for the SLM as-built Ti-2La alloy (Fig.1b and Fig. 2d, e) not showing (martensite) transformation from prior β grains.

Upon completion of the β → α transformation at 850 °C, the volume fractions of La-bcc and La-fcc start to decrease and increase, respectively (Fig. 4a). About 8.5vol.% of β, rich in La, remains in the microstructure at 850 °C. As temperature decreases, β rejects La as clearly revealed by the significant shift of {200}β towards {311}La-fcc in the temperature range of 850–550 °C. No evidence of an allotropic La-fcc → La-dhcp transformation indicated by the equilibrium phase diagram (Fig. 1a) is observed even for T < 400 °C at the investigated cooling rates.

The kinetics of α crystallization are compared for the cooling rates studied (5, 20, and 100 °C min$^{-1}$) in Supplementary Fig. 4a, b. They present an Avrami-like sigmoidal profile characterized by low transformation rates at the beginning and at the end of the β → α transformation, indicating a certain time required for nucleation and fall-off in the formation/growth of new grains, respectively[40]. Moreover, linear growth rates are obtained during the β → α transformation, characteristic of polymorphic crystallization where atom movements are essentially confined to the vicinity of the transformation front[41]. Supplementary Fig. 4c indicates that a decrease of −0.05% in the variation of the lattice parameter a$_β$ is obtained for all cooling conditions at the onset of the β → α transformation. The associated diffusion phenomena is a consequence of rejection of La from β to the β/L$_1$ interface[29], inducing a critical CU required to trigger extensive nucleation of α at the β/L$_1$ interface. This is reflected in the sudden and extensive formation of differently oriented α reflections at about 900 °C (Fig. 4b), while small α particles at former β/L$_1$ interfaces are identified in Fig. 4d. Thereafter, α grains grow into the primary β phase. Similarly to other peritectic systems[42,43], the results obtained for Ti-2La point to an important role of the

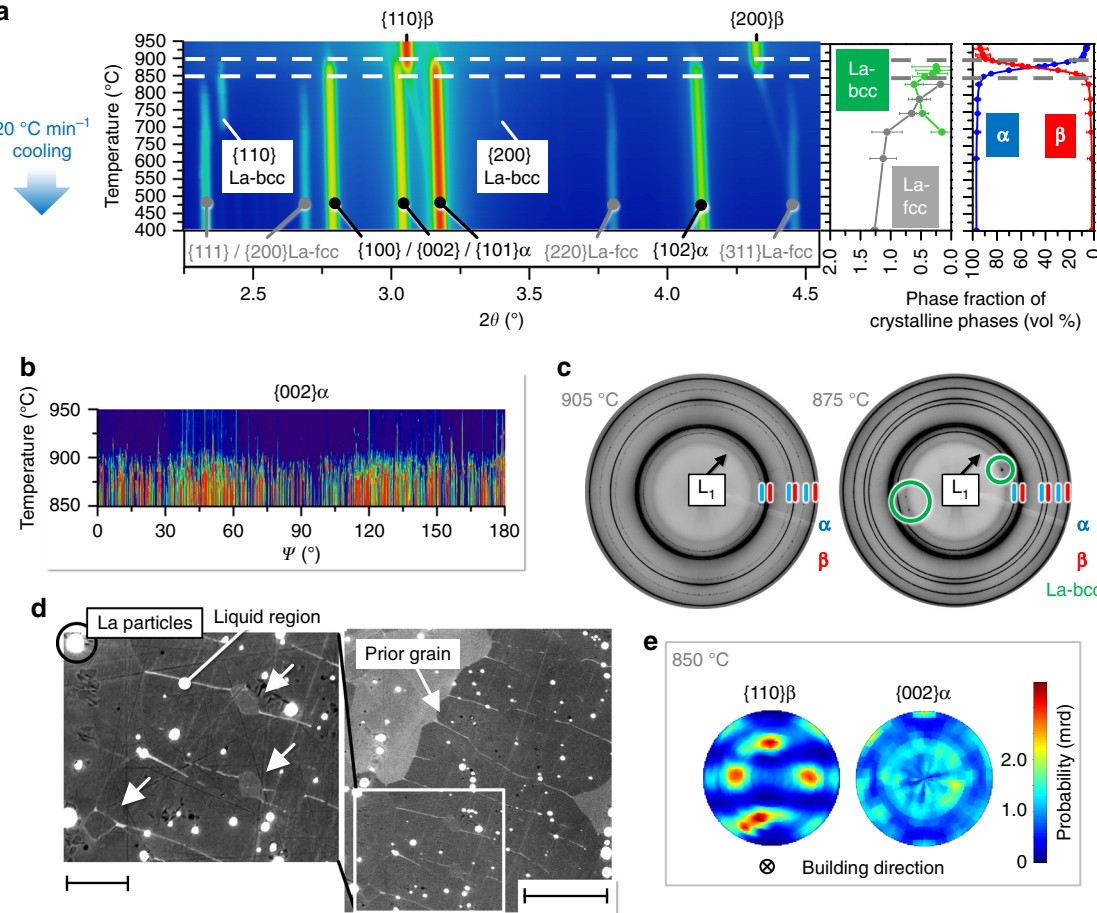

**Fig. 4** Phase transformation kinetics of the Ti-2wt.% La alloy. Formation of α starts in the L₁ + β field and, subsequently, α transforms from β during cooling. **a** Color-coded 2D plot of the evolution of {hkl} reflections of β, α, La-bcc and La-fcc for a representative 2θ range of 2.25–4.55°, combined with the simultaneous evolution of volume fractions of crystalline phases obtained from Rietveld analysis during continuous cooling from 950 °C down to 400 °C with 20 °C min⁻¹. No changes are observed for T < 400 °C. **b** Color-coded 2D plot of the evolution of {002}α Bragg reflections for an azimuthal angle (ψ) range 0–180° during continuous cooling between 950–850 °C with 20 °C min⁻¹. **c** Presence of α coexisting with β and L₁ phases during transformation is revealed in the complete Debye-Scherrer rings acquired at 905 °C and 875 °C. **d** Nucleation of α particles (pointed by arrows) at former β/L₁ interfaces is visible in a microstructure quenched from 950 °C. Scale bar, 5 μm (right side); 2 μm in magnified image (left side). **e** Normalized pole figures of {110}β and {002}α reconstructed from a gauge volume of 1 × 1 × 5 mm³, indicating that α does not inherit the texture of the parent β phase right after β → α transformation of Ti-2La during cooling down to 850 °C. The transformation of α from β (β → α) is reflected in the rapid increase in the volume fraction of α between 900–850 °C shown in **a**

growth restriction of the solute to induce CU providing the driving force for nucleation.

On the other hand, the grain refinement observed for Ti-2La with increasing cooling rate (Fig. 3), can be explained by a higher thermal undercooling provoking more nucleation sites. Coarser grains are obtained during slow cooling owing to more severe competition between nucleation and growth. The fast cooling rates reached during SLM can be exploited to generate fine-equiaxed microstructures as reflected in the SLM as-built condition (Fig. 1d, e).

**β-stabilizer addition**. The applicability of the peritectic phase transformation path discovered in this work for SLM Ti-2La, was explored as well for the ternary system Ti-Fe-La, i.e., by adding a β-stabilizing element. For this, Ti-3wt.% Fe (Ti-3Fe) and Ti-1.4Fe-1La (wt.%) were produced using the identical SLM strategies used for CP Ti and Ti-2La. To the best of the author's knowledge, no data about the ternary Ti-Fe-La phase diagram has been reported for the Ti-rich corner. The results shown in Fig. 5 reveal that additions of the peritectic forming element La can

induce as well an alternative path of α formation during SLM and post-thermal treatment of Ti-alloys. While SLM of Ti-3Fe results in a typical Widmanstätten microstructure via sympathetic β → α transformation (Fig. 5a), SLM of Ti-1.4Fe-1La leads to formation of a microstructure with α particles of diverse orientations together with α lamellae arranged in colonies (Fig. 5b, e). This microstructure presents a remarkable reduction of texture compared to Ti-3Fe, as inferred by comparing Fig. 5c, d. Upon post thermal treatment by slow cooling of these SLM as-built conditions from 950 °C with 20 °C min⁻¹ (i.e., from their corresponding β and L + β field down to RT), a typical lamellar α+β microstructure is obtained for Ti-3Fe (Fig. 5f), while Ti-1.4Fe-1La presents extensive globularization of α (Fig. 5g). This suggests that, similarly to Ti-2La, heterogeneous nucleation of α at solid/liquid interfaces also takes place in Ti-1.4Fe-1La. Accordingly, the bi-modal distribution of lattice correlation boundaries between α and β phases obtained by EBSD for this condition (Fig. 5h), shows the presence of two distinct paths of α formation: α formed via the Burgers OR and non-Burgers OR α derived from the path L₁ + β → α. Moreover, Supplementary Fig. 5 provides further crystallographic evidence of non-operating Burgers OR in the

**Table 1 Equilibrium invariant reactions according to the current knowledge of the equilibrium phase diagram[29,50] for the Ti-2wt. % La alloy (Ti-2La) contrasted with the transformations identified in situ (Fig. 4), using high-energy synchrotron X-ray diffraction (HEXRD) during cooling from 950 °C ($L_1 + \beta$ field) with 20 °C min$^{-1}$**

| Phase diagram | | In situ HEXRD | |
|---|---|---|---|
| Invariant reaction | Temperature (°C) | Suggested transformation | Temperature range (°C) |
| – | – | Peritectic $L_1 + \beta \rightarrow \alpha$ | 950–900 |
| – | – | Allotropic $\beta \rightarrow \alpha$ | 900–850 |
| Peritectic $L_1 + \beta \leftrightarrow$ La-bcc | 922 | Peritectic $L_1 + \beta \rightarrow$ La-bcc | 875–850 |
| Peritectoid La-bcc $+ \beta \leftrightarrow \alpha$ | 887 | – | – |
| Eutectoid La-bcc $\leftrightarrow$ La-fcc $+ \alpha$ | 857 | Eutectoid La-bcc $\rightarrow$ La-fcc $+ \alpha$ | 850–700 |
| Allotropic La-fcc $\leftrightarrow$ La-dhcp | 277 | – | – |

heat treated Ti-1.4Fe-1La alloy by considering adjacent α-β regions.

## Discussion

The results obtained in the present work, reveal an alternative path of α formation in titanium alloys by addition of the rare earth element La. This has been investigated after SLM, as well as during post-thermal treatment of Ti-2wt.% La and Ti-1.4-Fe-1La (wt.%) systems. α grains nucleate in the $L_1 + \beta$ field via peritectic $L_1 + \beta \rightarrow \alpha$ reaction and subsequently transform from β (β → α). The resulting α phase is not always related to the orientation of the parent β phase and therefore, significant texture reductions, as well as equiaxed microstructures can be achieved. The finding of this transformation path—to the best of our knowledge—of unprecedented exploitation in titanium alloys, opens up an alternative to avoid a deep-rooted drawback during AM of titanium alloys, namely epitaxial growth generating coarse columnar β grains with strong <100> β orientation along the building direction.

The approach of adding peritectic forming elements capable to induce as-built, as well as post-processed additive-layer-manufactured microstructures of reduced texture, can have a positive impact in commercial titanium compositions. Moreover, the general idea of adapting alloys to AM using a peritectic reaction can open up windows for target oriented alloy design in other alloy systems. This will be the subject of future investigations.

## Methods

**Selective laser melting.** Selective laser melting (SLM) of powders of commercially pure titanium grade 1 (CP Ti, with max. 0.18 wt.% O and 0.2 wt.% Fe), as well as of powder blends of CP Ti-2wt.% La and CP Ti-1.4Fe-1La (wt.%) was carried out with flowing argon 5.0 (i.e., purity >99.999%) at a consumption rate of ~2.5 L min$^{-1}$. A SLM 280HL machine incorporating an in-situ melt pool-monitoring system that can detect hot/cold spots during SLM was employed. The building platform was not externally heated during SLM processing. A reduced build envelope of 50 × 50 × 50 mm$^3$ was used. The boiling point, $t_b$, of Ti, La, and Fe is 3287, 3464, and 2861 °C, respectively[44]. Compared with the classical Ti-6Al-4V (with Al, $t_b$ = 2519 °C), a less critical role of element evaporation is expected for the studied CP Ti, Ti-2La, and Ti-1.4Fe-1La alloys under the same processing conditions.

The temperature of the building platform made of Ti-6Al-4V (wt.%) and the oxygen content were maintained <40 °C and <0.14%, respectively. The alloys were produced by blending the base powder of CP Ti with additions of commercial powder of pure La (99.9%) and Fe (>99%) elements with maximal and mean particle sizes of ~74 μm and ~3.5 μm, respectively. Powder blending was performed inside stainless steel containers within a glovebox kept in an atmosphere of argon 5.0 of purity and <1 ppm oxygen content. Thereafter, flowability tests were successfully carried out employing a stand Ti-6Al-4V funnel for testing free-flowing metal powder according to ISO 4490:2001. The containers—of ~ 12 kg of powder capacity—were tightly sealed by using a valve inside the glovebox. By doing so, they were prepared to be placed in the SLM machine.

The SLM equipment was supplied by the SLM solutions GmbH, CP Ti powder was produced by gas atomization by TLS Technik GmbH, La (packaged in Ar atmosphere), and Fe powders were produced by Alfa Aesar and BASF CEP SM, respectively. The cost of La powder was 23 € g$^{-1}$. However, the price of La strongly varies depending on the element form (e.g., powder compared to pieces). For instance,

it can be significantly cheaper than V, alloying element of the most commercialized Ti alloy Ti-6Al-4V[45]. CP Ti powder—with β-transus temperature of ~890 °C—consisted of spherical particles with a size distribution according to the following D-values (CILAS 990 NASS): D10 = 28 μm, D50 = 39 μm and D90 = 51 μm.

Seven cubes of 10 × 10 × 10 mm$^3$ were built on support structures (height = 1 mm) in a single SLM manufacturing process using a checkerboard scanning with an increment of 90° from layer-to-layer. Three SLM manufacturing jobs were carried out independently for CP Ti, Ti-2La, and Ti-1.4Fe-1La. These sample dimensions permit evaluating SLM synthetized microstructures from different powder blends. The following are the main SLM processing parameters employed for the studied alloys: laser power = 350 W, scanning velocity = 1000 mm s$^{-1}$, hatch distance = 40 μm, focal offset distance = 0 mm, and layer thickness = 50 μm, resulting in a volume energy density = 175 J mm$^{-3}$. This strategy aims at applying a severe intrinsic heat treatment induced by the scanning laser[6] in order to promote intensive re-melting of solute and CP Ti powders, i.e., increase the compositional homogeneity of the manufactured as-built bulk alloys. The chosen SLM parameters promoted minimization of irregular distributions of solute La. Thus, the main focus of the SLM process was reaching extensive in situ alloying of powders in a single SLM process step. The delay time between two subsequent layers was 120 s.

**Microscopy.** Metallographic examinations of the central section of as-built SLM cubes (parallel to the building direction), were performed using light optical microscopy (LOM), scanning electron microscopy (SEM) in backscattered electron mode (BSE) and electron backscatter diffraction (EBSD) analysis. The specimens were prepared by grinding and polishing (3 μm diamond suspension and SiO$_2$-H$_2$O-H$_2$O$_2$ for the last two steps) using a TegraPol machine. The samples for LOM were immersed for 1 min in a 3% HF reagent solution at room temperature to reveal the microstructural constituents. LOM and SEM were carried out using a Zeiss LSM 700 laser scanning microscope in visible light mode and a dual beam FEI Helios Nanolab 600i (electron and Ga+) setup with an integrated SEM unit, respectively.

For EBSD, a Zeiss Ultra 55 field emission gun scanning electron microscope (FEG-SEM) operated at 20 kV was employed in combination with an Oxford Nordlys II EBSD detection system using the acquisition software AZtec (Version 3.3). The beam step-size, working distance and sample-tilt angle used were 2.2 μm and 0.05 μm, 10 mm and 70°, respectively. Post-processing of EBSD data was carried out with the software Channel 5. Distribution of lattice correlation boundaries between α and β phases was performed for an area of 350 × 250 μm$^2$. Reconstruction of parent β grains from EBSD phase orientation maps was performed using the software ARPGE[30].

**Hardness.** Vickers microhardness measurements along the mirror polished, unetched central section of as-built cubes parallel to the SLM building direction, $z$, were performed using a Clemex MMT-X7 tester. Line sequences of indentations covering the complete height of the sample (from $z = 0$ mm to $z = 10$ mm) were carried out using a force of 200 g (HV$_{0.2}$). The values obtained correspond to an average of 110 different indentations taken along $z$. The size of the resulting indentations is considerably larger than that of the microstructural features.

**In situ high energy synchrotron X-ray diffraction.** In situ HEXRD was carried out at the P07-HEMS beamline of PETRA III (Deutsches Elektronen-Synchrotron) [46] using an energy of 100 keV (λ = 0.124 Å). Patterns of entire Debye-Scherrer rings were acquired ex-situ from the bulk of SLM as-built samples, as well as in situ during subsequent thermal treatment of SLM as-built conditions in vacuum atmosphere. For that, a modified dilatometer Bähr 805 A/D equipped with an induction coil furnace was used in combination with a PerkinElmer XRD 1621 image plate detector[47]. This dilatometer was employed to perform argon quenching of samples for subsequent metallographic analysis. The acquisition time and sample-detector distance were 4 s and 1645 mm, respectively. Polished samples of 5 × 5 × 10 mm$^3$ (thickness = 5 mm) cut with a SiC disk from the center of as-built SLM cubes, were investigated and kept fix during acquisition. The

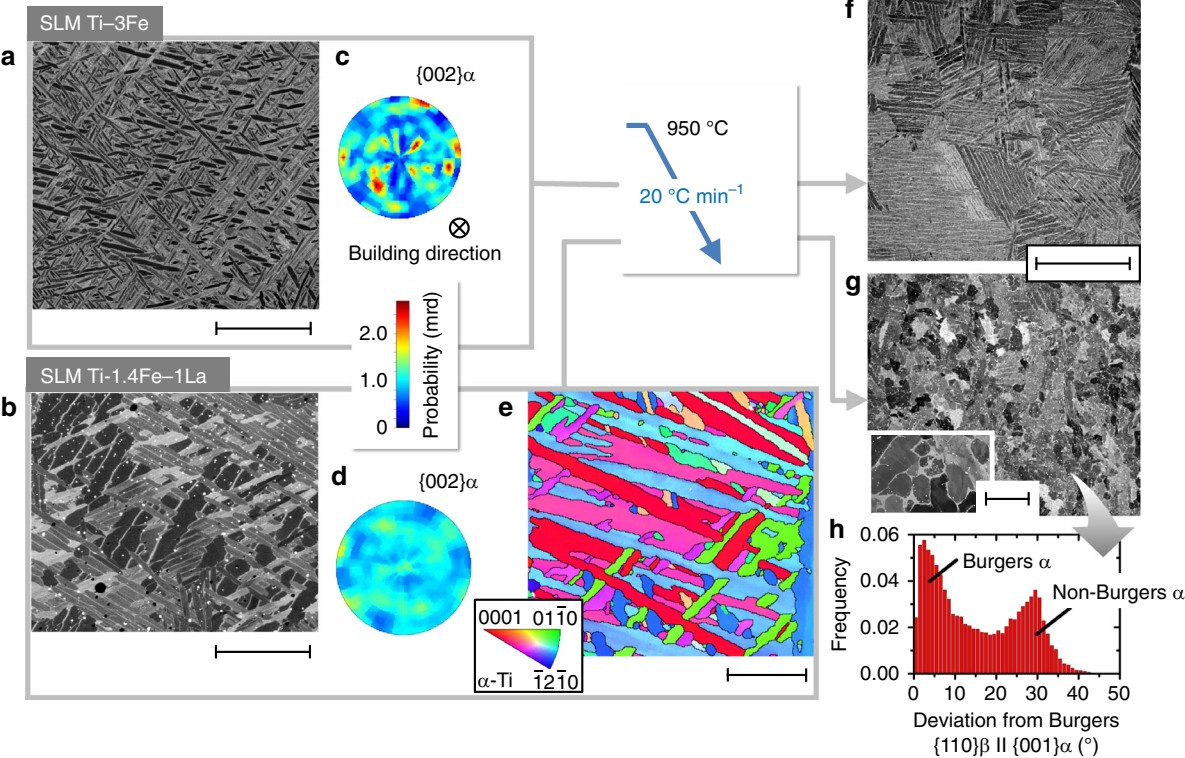

**Fig. 5** Reducing solidification texture during additive manufacturing of titanium alloys. Formation of fine α particles with multiple orientations occurs within colonies of α lamellae by addition of the peritectic forming solute La. **a**, **b** Transformation of α during selective laser melting (SLM) of Ti-3wt.% Fe (Ti-3Fe) and Ti-1.4Fe-1La (wt.%), respectively (scale bars, 10 μm). For the first case, α forms directly from parent β grains: the typical path of β → α transformation that results in a lamellar α+β microstructure. For the second, an additional path of α formation not linked with the orientation relationship of the parent β phase takes place. This is reflected in **c**, **d**: normalized pole figures of {002}α reconstructed from a gauge volume of 1 × 1 × 5 mm³ for Ti-3Fe and Ti-1.4Fe-1La, respectively. **e** α particles of diverse orientations that can reach diameters <1 μm form at boundaries of α lamellae during SLM of Ti-1.4Fe-1La. Black lines indicate high angle grain boundaries (misorientation >10°). Scale bar, 5 μm. **f**, **g** Post thermal treatment by slow cooling the Ti-3Fe and Ti-1.4Fe-1La alloys from 950 °C with 20 °C min⁻¹ down to room temperature, leads on the one hand, to a typical lamellar α + β microstructure and on the other, to extensive globularization of α, respectively. Scale bars in **f** and **g**, 250 μm; 30 μm in the inset of **g**. **h** The two paths of α formation taking place for the heat treated Ti-1.4Fe-1La alloy are reflected in the bi-modal distribution obtained for lattice correlation boundaries between α and β phases

temperature was controlled by a spot-welded thermocouple located next to the position of the incident beam of slit size = 1 × 1 mm², i.e., close to the center of the building height of the specimens. The investigated CP Ti, Ti-2La, and Ti-1.4Fe-1La alloys were probed by in situ HEXRD using a gauge volume of 1 × 1 × 5 mm³.

Qualitative analysis of the evolution of the diffraction patterns during thermal treatment was carried out by converting Debye-Scherrer rings into Cartesian coordinates (Azimuthal angle ψ, 2θ). Subsequently, projection of the summed intensity of Bragg reflections on the 2θ and Azimuthal axis was performed using the software ImageJ[48]. Quantitative phase analysis of the diffraction patterns was carried out using the Rietveld method as implemented in the software Maud[49]. An E-WIMV algorithm integrated in this software was used for texture analysis. The instrumental parameters of the HEXRD setup were obtained using a LaB₆ powder standard.

**Data availability**. The data that support the findings of this study are available from the corresponding author upon reasonable request.

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

## Acknowledgements

The Deutsches Elektronen-Synchrotron (DESY) is acknowledged for the provision of synchrotron radiation facilities in the framework of the proposal I-20170438. The authors would like to thank T. Merzouk and P. Watermeyer for their support during SLM manufacturing and metallographic investigations, respectively.

## Author contributions

P.B.-V., J.H., J.G., and G.R. conceived and designed the project. P.B.-V., J.H., J.G., G.R., A.S., and N.S. performed the synchrotron experiments. P.B.-V. and J.G. performed all other remaining experiments. P.B.-V. analyzed the data. P.B.-V., J.H., J.G., and G.R. interpreted, discussed the results and wrote the manuscript.

## Additional information

**Competing interests:** The authors declare no competing interests.

