## [Peer Review File · Nature Communications]

Reviewers' comments:

Reviewer #1 (Remarks to the Author):

The authors present a rather unusual binary (and later, by extension, ternary) titanium-based alloy system in which the microstructure is observed to be more equiaxed in nature following additive manufacturing. The Ti-La system is generally a under explored binary system.

The general concept of breaking up texture in additively manufactured titanium alloys is of broad interest to the community, and has been the subject of several papers over the past few years, and related to grain size in titanium alloys in general. In particular, the previous use of boron has been reported to have a strong influence on reducing both texture (as represented by the multiple times random distribution of the orientation data) and grain size. The authors are encouraged to consider adding a reference to this work (e.g., "The effect of boron on the grain size and texture in additively manufactured β -Ti alloys." *Journal of Materials Science* 52, no. 20 (2017): 12455-12466.). This is by no means a requirement, but the authors may find some interesting future opportunities, as there has been speculation in the literature about whether boron disrupts the Burgers orientation relationship. The reviewer is aware of other work that has investigated additions of Fe as well.

What is most interesting is the control of texture through the peritectic reaction. The authors have investigated a seemingly unique space, and have added to the overall field. I feel that the authors, and others in the community, may be entering a fairly interesting period of time when questions of solidification pathways are discussed and debated. I am looking forward to seeing how the fundamental science unfolds.

The evidence for the elimination of the Burgers Orientation Relationship lacks some robustness. It adopts more of a "statistical" approach rather than a rigorous crystallographic approach. The paper would be greatly enhanced by TEM or high resolution EBSD evidence from adjacent alpha/beta regions showing that the Burgers OR is not operating across many different locations (at least 2-3). The authors may be able to get such information from the Ti-1.4Fe-1La system (e.g., material in fig. 5g analyzed by EBSD, as there is a sufficient volume fraction beta).

Here are a few comments that the authors may consider which may improve their overall paper:

* (Lines 51-59) The authors refer to martensitic structures in titanium as "brittle". I concur that generally they may be less ductile than pure titanium or many alloys. However, this is not intrinsically the same as "brittle" in the martensitic transformation in steels, as the crystal structure is simply a supersaturated hcp (or orthorhombic). Any brittleness in additively manufactured components exhibiting martensite may well be associated with other defects, and not simply the martensite. The martensite and brittleness is not the subject of the rest of the paper (e.g., no mechanical properties for the Ti-La depositions are presented), so the authors may want to consider whether this section could be reduced or eliminated. Within the same section, lines 56-68 refer to post-thermal processes. If this overall section is kept, the authors should add a reference for post-thermal and/or thermomechanical processes.

* Line 87: "...with the c-axis slightly tilted with respect...". This should be quantified. The authors may find it closer to 30-40°.

* Line 92: The authors assume a Burgers OR holds for CP. This is not an unreasonable assumption, as the Burgers OR has been repeatedly established as the dominant OR for titanium beta-to-alpha phase transformations. However, later the authors assume that the Burgers OR does not occur for the Ti-2La system. It is the latter one which is presented without traditional evidence which presents a gap (Lines 157-158).

* Line 98-100: There are papers in the literature which refer to the columnar-to-equiaxed transition (CET), in which similar bands of fine equiaxed grains followed by columnar grains are observed. The authors might want to consider whether this phenomenon is present, and if not, might point out that this is not a traditional CET. If the authors are unsure, they may simply refer

that one such explanation is the CET (in addition to "the chosen SLM strategy").

* Line 126: The authors use a rather ambiguous term "rapid". Can this be quantified with respect to time?

* Line 137: 5100°C/min - I believe there is a typo here.

* Line 153: Suggest replacing "irrelevant" with "negligible"

* Lines 157-158: The authors use pole figures and multiple of random distribution (MRD) values to underpin their argument that the Burgers OR is not adopted. There is insufficient evidence that this is true. E.g., Fig. 2 (a,b) show a mrd of 2.4 and 1.9, with hot spots in similar locations. Arguably, this may be a 30-35% reduction in texture, but clearly the Burgers OR is still active. Traditional evidence would include TEM microscopy (or higher resolution EBSD), and ultimately, the authors would be more able to convincingly prove that this is true.

* Lines 174 and figures: Comments on tortuous primary alpha grains. This is an interesting result. The authors are encouraged to vary the input energy to eliminate low input energies which, when coupled with elemental blends, cause challenges and local "cold spots" in the liquid as the elements separately melt and then have the thermodynamics such as the enthalpy of mixing come into play.

* Line 200: The authors use the short-hand "CU" for constitutional undercooling, yet this the only time this abbreviation appears. The authors should simply use "constitutional undercooling", as there is no need for an abbreviation for one use.

* Line 243: The authors speculate that Mg or Ca could be used, but each of these present challenges given their melting point.

* Line 249: What does "argon 5.0 atmosphere employing" mean? Does this mean it was a positive pressure of 5x atmosphere?

Reviewer #2 (Remarks to the Author):

The authors report on a new Ti-base alloy for additive manufacturing (AM, also referred to as 3D printing). Even if AM already is used in industry for realization for complex components, numerous upon questions/issues remain.

One major roadblock to more widespread applications is the limited number of alloys being available to be processed employing powder bed techniques such as selective laser melting (SLM). Most importantly, alloys employed so far have been developed for conventional processing routes, e.g. forging and casting. Thus, these alloys are not adapted to the unique processing conditions prevailing in AM and SLM, respectively. Rapid solidification and epitaxial growth are two aspects to be mentioned in this regard. In the field of Ti-base alloys, almost exclusively Ti-6Al-4V and commercial purity (CP) titanium are considered in both academia and industry. It is well accepted that these alloys suffer from several process-induced issues: anisotropic microstructure and mechanical properties as well as low damage tolerance. Thus, post treatments always need to be conducted. Still, the anisotropic nature of deformation prevails upon standard post processing routes.

In consequence, development of new Ti-alloys meeting the process conditions of AM and SLM, respectively, and allowing for realization of isotropic microstructures are crucially needed for further development in the field. By adding La to CP titanium the authors address this topic in an excellent way. Results obtained are of highest quality and were elaborated using absolutely sophisticated characterization techniques, e.g. high-energy X-ray diffraction (HEXRD). Conclusions drawn based on data presented are fully convincing.

The general idea of adapting Ti-alloys to AM by using a peritectic reaction is absolutely novel and will open up windows for target oriented alloy design in other alloy systems as well.

The thorough experimental effort, quality of data, in-depth discussion and expected impact of the approach presented are clearly up to the standard of Nature Communications.

Quality of figures and text and, thus, presentation of results is excellent.

In consequence, the reviewer strongly recommends acceptance of the current work.

In order to further strengthen their contribution, the authors should consider the following (not mandatory):

In the introduction section, numbers provided should be substantiated. Production savings up to 50% are highlighted. Which kind of components are referred to here?

Furthermore, additional references to current literature could be provided. Realization of isotropic microstructures in Al-alloys as well as steels has been published quite recently: use of nano-sized particles for grain refinement in high-strength aluminium alloys and employment of multiple phase transformations induced by intrinsic heat treatment in high alloyed steels. These concepts should be introduced shortly in the introduction section even if not being applicable to Ti-base alloys so far.

The authors should plot the binary phase diagram in a way that allows highlighting eutectic, peritectoid and eutectoid reaction in the system.

Cost for the element La should be provided.

Some details mentioned in the text (e.g. agglomerations of α' plates (page 5, line 98)) cannot directly be seen in the corresponding figures. The authors should add further markers to highlight these features.

The authors should consider showing (part of) the phase diagram for the ternary Ti-Fe-La system in the supplement.

When discussing results shown (or in the corresponding part of the methods section) clear statements on number of samples and sample volume probed have to be provided.

Using powder blends for AM always opens up the following questions: What are the characteristics of all powders employed (details on La and Fe powders are missing). Using which apparatus the powders were mixed? Are all samples processed homogeneous in terms of chemical composition? Regarding SLM processing the following questions should be answered: Is any information on element evaporation available? What size of build envelope was employed? Is there any information on gas flow?

Scale bars in Figure 2 remain unclear: Scale bars of 100 microns and 50 microns seem to be valid for c and d, respectively. The overall size of both figures is very similar. The boxes highlighting the areas depicted in high resolution at the bottom are clearly different in size in the overview images. Thus, the same scale bar cannot be correct for c and d. Please clarify.

Finally, showing a stress-strain curve from tensile testing would be very helpful to draw a complete picture of the new material processed. Are data available?

Reviewer #3 (Remarks to the Author):

This is a well-written paper with an interesting result, namely that La can be useful for changing the solidification (phase) path in titanium. I appreciate the use of synchrotron x-ray diffraction experiments to complement the microstructural characterization. Unfortunately, justifying publication in Nature Communications on the basis of mitigating strong texture in typical additively printed Ti-6Al-4V is not supported by the literature because, in fact, the texture in the dominant (~90 %) alpha phase is weak; see, e.g., A. A. Antonysamy, J. Meyer, and P. B. Prangnell, Effect of build geometry on the beta-grain structure and texture in additive manufacture of Ti-6Al-4V by selective electron beam melting, *Materials Characterization*, 84, 153-168 (2013). That the texture is weak is unsurprising because when the strongly columnar (bcc) beta transforms to (hexagonal) alpha following the Burgers orientation relationship, the high driving force available under rapid cooling means that there is negligible variant selection and the texture in the alpha is weak. Also, martensitic structures can be easily reverted to two-phase via heat treatment which is generally done if for no other reason than stress relief, especially in laser powder bed materials. A second criticism concerns the interpretation of the tortuous grain shapes that were observed: the authors need to read the literature on 3D printing and check what scan strategy was used in their particular builds because it can happen that offsets and changes in direction force the direction of

the thermal gradient (and therefore the solidification path) to change substantially from layer to layer, resulting in the observed pattern. By contrast, scanning in the same direction in successive layers with no offset in lateral position can result in very strong epitaxial growth. This means that texture is at least as much influenced by scan strategy as by local solidification conditions. I apologize for not offering more detailed feedback but the authors need to re-think the basis for the originality of their work.

1. Reviewer #1 (Remarks to the Author):

C1. *The authors present a rather unusual binary (and later, by extension, ternary) titanium-based alloy system in which the microstructure is observed to be more equiaxed in nature following additive manufacturing. The Ti-La system is generally a under explored binary system.*

*The general concept of breaking up texture in additively manufactured titanium alloys is of broad interest to the community, and has been the subject of several papers over the past few years, and related to grain size in titanium alloys in general. In particular, the previous use of boron has been reported to have a strong influence on reducing both texture (as represented by the multiple times random distribution of the orientation data) and grain size. The authors are encouraged to consider adding a reference to this work (e.g., "The effect of boron on the grain size and texture in additively manufactured β -Ti alloys." *Journal of Materials Science* 52, no. 20 (2017): 12455-12466.). This is by no means a requirement, but the authors may find some interesting future opportunities, as there has been speculation in the literature about whether boron disrupts the Burgers orientation relationship. The reviewer is aware of other work that has investigated additions of Fe as well.*

Indeed, previous investigations modified the chemical composition of titanium alloys for additive manufacturing (AM) by addition of B and Fe. Similarly to the problematic presented in our work, these approaches aim at tackling epitaxial growth in AM-produced titanium alloys, reporting remarkable

advances. The authors acknowledge the contribution of the reviewer, and would like to cordially refer her/him to the introduction of the resubmitted manuscript. There, a new paragraph has been included considering the citation suggested¹⁹, in addition to four additional complementary works^{20–23} that provide a substantial contextualization framework:

19. Mantri, S.A. et al. The effect of boron on the grain size and texture in additively manufactured β -Ti alloys. *J. Mater. Sci.* **52**, 12455–12466 (2017).

20. Bermingham, M.J., Kent, D., Zhan, H., StJohn, D.H. & Dargusch, M.S. Controlling the microstructure and properties of wire arc additive manufactured Ti–6Al–4V with trace boron additions. *Acta Mater.* **91**, 289–303 (2015).

21. Banerjee, R., Collins, P.C., Genc, A. & Fraser, H.L. Direct laser deposition of in situ Ti-6Al-4V-TiB composites. *Mater. Sci. Eng. A* **358**, 343–349 (2003).

22. Winstone, M.R., *Titanium Alloys at Elevated Temperature: Structural Development and Service Behaviour* 165–175 (IOM Communications, 2001).

23. Azizi, H. et al. Additive manufacturing of a novel Ti-Al-V-Fe alloy using selective laser melting. *Addit. Manuf.* **21**, 529–535 (2018).

The changes described in the previous lines can be found in the new version of the manuscript:

Page 5, lines 84–96:

“Approaches to tackle epitaxial growth in AM titanium alloys include addition of B to $\alpha+\beta$ and β compositions. For instance, the effect of B on powder blends of Ti-20V, Ti-12Mo¹⁹, and on the Ti-6Al-4V powder alloy^{20,21} can result in grain refinement. Globularization of the microstructure preserving the Burgers orientation relationship (OR) between α and β phases has also been reported¹⁹. The use of B leads to formation of ceramic TiB-needles. Thus, the presence of TiB has been associated with strength increase at expenses of ductility as well as localized plastic flow and damage caused by inhomogeneous distributions of the TiB needles^{20,22}. Other investigations with a Ti-1Al-8V-5Fe alloy produced by SLM showed globularization and formation of small β grains along the building direction owing to the presence of Fe²³. In addition to inhomogeneous distribution of α laths resulting from partitioning and segregation of Fe, the as-built microstructure obtained was formed by α films along β grain boundaries which are detrimental to fatigue and ductility in β Ti-alloys²⁴.

C2. *What is most interesting is the control of texture through the peritectic reaction. The authors have investigated a seemingly unique space, and have added to the overall field. I feel that the authors, and others in the community, may be entering a fairly interesting period of time when questions of solidification pathways are discussed and debated. I am looking forward to seeing how the fundamental science unfolds.*

The evidence for the elimination of the Burgers Orientation Relationship lacks some robustness. It adopts more of a "statistical" approach rather than a rigorous crystallographic approach. The paper would be greatly enhanced by TEM or high resolution EBSD evidence from adjacent alpha/beta regions showing that the Burgers OR is not

operating across many different locations (at least 2-3). The authors may be able to get such information from the Ti-1.4Fe-1La system (e.g., material in fig. 5g analyzed by EBSD, as there is a sufficient volume fraction beta).

The new figure “Supplementary Fig. 5” has been included in the resubmitted version of the manuscript in order to provide robustness to the evidence of the non-Burgers orientation relationship in the Ti-1.4Fe-1La system. New EBSD analyses have been performed. Following the reviewer’s suggestion, evidence of non-operating Burgers relationship is now provided for 3 different locations considering adjacent α - β regions. Moreover, this effect is also presented for the complete studied area:

Supplementary Fig. 5 Post-thermal treatment by slow cooling the SLM Ti-1.4Fe-1La alloy from 950°C with 20°C/min down to room temperature leads to extensive globularization of α . **a** EBSD phase map for α and β (in blue and red, respectively). IPF maps of β and the corresponding pole figures for $\{110\}\beta$ as well as $\{001\}\alpha$ obtained for the α grains pointed in **a**: **b** Grain 1, **c** Grain 2, **d** Grain 3. The α variants (encircled in red) do not present the usual Burgers OR $\{001\}\alpha \parallel \{110\}\beta$ with respect to its surrounding β matrix (in blue). This can also be observed when considering the complete area shown in **a**, where **e** encircled regions of $\{001\}\alpha$ variants (in red) do not match those of $\{110\}\beta$ (in blue). This reflects the non-Burgers path of α formation taking place for the heat treated Ti-1.4Fe-1La alloy.

The modifications described in the previous lines have been clarified in the resubmitted manuscript:

Page 14, lines 304–310:

“Accordingly, the bi-modal distribution of lattice correlation boundaries between α and β phases obtained by EBSD for this condition (Fig. 5h), shows the presence of two distinct paths of α formation: α formed via the Burgers OR and non-Burgers OR α derived from the path $L_1+\beta \rightarrow \alpha$. Moreover, Supplementary Fig. 5 provides further crystallographic evidence of non-operating Burgers relationship in the heat treated Ti-1.4Fe-1La alloy by considering adjacent α - β regions.”

C3. *Here are a few comments that the authors may consider which may improve their overall paper:*

** (Lines 51-59) The authors refer to martensitic structures in titanium as "brittle". I concur that generally they may be less ductile than pure titanium or many alloys. However, this is not intrinsically the same as "brittle" in the martensitic transformation in steels, as the crystal structure is simply a supersaturated hcp (or orthorhombic). Any brittleness in additively manufactured components exhibiting martensite may well be associated with other defects, and not simply the martensite. The martensite and brittleness is not the subject of the rest of the paper (e.g., no mechanical properties for the Ti-La depositions are presented), so the authors may want to consider whether this section could be reduced or eliminated. Within the same section, lines 56-68 refer to post-thermal processes. If this overall section is kept, the authors should add a reference for post-thermal and/or thermomechanical processes.*

The authors agree with the points raised by the reviewer. Following her/his suggestion, a broader explanation is included in the new version of the text clarifying the different causes of brittleness occurring in steels and titanium alloys containing martensite. As this phase is usually obtained upon AM of $\alpha+\beta$ titanium alloys, providing poor ductility and fracture toughness, this additional description requested in the introduction will offer a better understanding of the goal pursued by our investigations. Moreover, the lack of specification referencing the post processing of martensitic microstructures has been corrected in the resubmitted version, supported by the following new key references:

11. Xu, W., Lui, E.W., Pateras, A., Qian, M. & Brandt, M. In situ tailoring microstructure in additively manufactured Ti-6Al-4V for superior mechanical performance. *Acta Mater.* **125**, 390–400 (2017).
12. Yang, J. et al. Formation and control of martensite in Ti-6Al-4V alloy produced by selective laser melting. *Mater. Des.* **108**, 308–318 (2016).
13. Berns, H., Theisen, W. *Ferrous materials: steel and cast iron* (Springer, Leipzig, 2008).
14. Vilaro, T., Colin, C. & Bartout, J.D. As-fabricated and heat-treated microstructures of the Ti-6Al-4V alloy processed by selective laser melting. *Metal. Mater. Trans. A* **42**, 3190–3199 (2011).
15. Popov, V. et al. Effect of hot isostatic pressure treatment on the electron-beam melted Ti-6Al-4V specimens. *Procedia Manuf.* **21**, 125–132 (2018).

The modifications requested resulted in the incorporation of the following new text:

Page 4, lines 67–77:

“Though in Ti-alloys both α' and the stable α phase present a hexagonal close-packed (hcp) lattice, the low ductility (< 10%) and fracture toughness exhibited by martensitic microstructures upon AM manufacturing is mainly a consequence of a high density of defects (e.g. dislocations, twins) present in the α' phase^{11,12}.”

Differently, the brittleness resulting from martensite formation in steels is associated with the distorted bcc tetragonal lattice containing ordered arrangements of interstitial C atoms¹³.

Post-thermal and/or thermomechanical treatments are commonly applied to as-built AM components to improve the strength-ductility trade-off. This can include supertransus or subtransus heat treatments¹⁴, as well as hot isostatic pressing¹⁵ where formation of stable α and β is induced via decomposition of metastable microstructures.”

C4. * *Line 87: "...with the c-axis slightly tilted with respect...". This should be quantified. The authors may find it closer to 30-40°.*

The tilt of the *c*-axis of the hcp lattice with respect to the SLM building direction (Fig. 2) has been quantified from the stereographic projections and found to be between 42.4–49°. These values are close to the Burgers orientation relationship (OR) $\{002\}_\alpha \parallel \{110\}_\beta$ of 45° between the hcp (α phase) and bcc lattices (β phase), when a strong $\langle 100 \rangle_\beta$ orientation occurs along the building direction. This new information has been introduced in the text:

Page 7, lines 134–138:

“(Fig. 2a,b) reveal a higher preferential orientation of the hcp lattice of CP Ti with the *c*-axis slightly tilted between 42.4–49° with respect to the SLM building direction. These values are close to the 45° expected from the Burgers OR $\{002\}_\alpha \parallel \{110\}_\beta$ when preferential $\langle 100 \rangle_\beta$ orientation occurs along the building direction³⁰.”

C5. * *Line 92: The authors assume a Burgers OR holds for CP. This is not an unreasonable assumption, as the Burgers OR has been repeatedly established as the dominant OR for titanium beta-to-alpha phase transformations. However, later the authors assume that the Burgers OR does not occur for the Ti-2La system. It is the latter one which is presented without traditional evidence which presents a gap (Lines 157-158).*

A new figure, “Supplementary Fig. 3”, has been included in the resubmitted version of the manuscript with the goal of providing crystallographic evidence of the non-occurrence of the Burgers OR in the Ti-2La system. This is supported by new EBSD analyses included in this figure for three different adjacent α - β regions:

Supplementary Fig. 3 Post-thermal treatment of the SLM Ti-2La as-built condition via slow cooling with 20°C/min from 950°C passing through the peritectic line (i.e. from $L_1+\beta$ field down to room temperature), provokes formation of α grains and extensive globularization. **a** EBSD phase map for α and β (in blue and red, respectively). The microstructure mainly consists in α , though retained β remains in triple junction boundaries (~1.4vol.% according to HEXRD). EBSD phase maps from three sub-regions of **a** (**b** triple junction 1, **c** triple junction 2, **d** triple junction 3) are presented with its corresponding IPF maps and pole figures for $\{110\}\beta$ as well as $\{001\}\alpha$. The α variants (encircled in red) surrounding β particles (in blue) do not present a usual Burgers OR $\{001\}\alpha \parallel \{110\}\beta$. This reflects the non-Burgers path of α formation taking place for the heat treated Ti-2La alloy. Black lines in EBSD phase maps **a**, **b**, **c** and **d** indicate high angle grain boundaries (misorientation $>10^\circ$).

The changes incorporated are reflected in the manuscript's new version:

Page 11, lines 227–234:

“The pole figures in Fig. 4e for $\{002\}\alpha$ and $\{110\}\beta$ at 850°C, i.e. at the end of the $\beta \rightarrow \alpha$ transformation, show that α forms to a large extent with orientations distinct from those given by the strong poles shown by β . This indicates that a fraction of α does not inherit the texture of β . This, together with local crystallographic orientation analyses of adjacent α - β grains of the condition obtained upon cooling down to RT (Supplementary Fig.3), indicates that the usual Burgers OR relationship $\{002\}\alpha \parallel \{110\}\beta$ typical for $\beta \rightarrow \alpha$ transformation is partially avoided.”

C6. * Line 98-100: There are papers in the literature which refer to the columnar-to-equiaxed transition (CET), in which similar bands of fine equiaxed grains followed by columnar grains are observed. The authors might want to consider whether this phenomenon is present, and if not, might point out that this is not a traditional CET. If the authors are unsure, they may simply refer that one such explanation is the CET (in addition to "the chosen SLM strategy").

The reviewer brings a meaningful aspect for complementing the discussion in our investigations upon selective laser melting (SLM) of Ti-2La. Accordingly, we believe that two reasons may influence the microstructure observed in Fig. 2d: constitutional undercooling (CU) and the chosen laser scanning strategy. The former can provide the driving force for nucleation leading to equiaxed microstructures. In the previous version of the manuscript, the influence of CU on the formation of α was focused on post-thermal treatment, since this condition leads to extensive formation of equiaxed α grains, and permits evaluation of this phenomenon in the basis of experimental in-situ / ex-situ results (see page 13, lines 269–276).

In the resubmitted text, the possible role of CU on CET occurring during SLM has been included. For instance, CU can vary in heat-affected regions promoting formation of fine-equiaxed α grains instead of columnar growth. Re-melting and re-heating of previously deposited material may improve the compositional homogeneity of the Ti-2La powder blending, and also create new nuclei via CU during this local cyclic remelting of the alloy. The low solubility of La in α -Ti and β -Ti, may as well contribute to a large CU. Though columnar grain growth is observed, fine-equiaxed α grains ($<10\mu\text{m}$) coexist with the elongated and tortuous α grains suggesting nucleation ahead of the solidification front, i.e. mixed solidification modes (columnar and equiaxed) may be acting in these regions (Fig. 2e). In addition to the CU effect commented, the microstructure of Fig. 2d is also a result of the chequerboard SLM strategy since the scanning direction of the laser, i.e. the thermal gradient front of the melt pool, changes for each layer.

The previous description, providing further insights into microstructure formation, has been included in the manuscript supported by two new references:

31. Kobryn, P. & Semiatin, S. Microstructure and texture evolution during solidification processing of Ti–6Al–4V. *J. Mater. Process. Technol.* **135**, 330–39 (2003).

32. Dehoff, R.R. et al. Site specific control of crystallographic grain orientation through electron beam additive manufacturing. *Mater. Sci. Technol.* **31**, 931–38 (2015).

Page 7, lines 152–172:

“Columnar-to-equiaxed transition (CET) of grain formation has been related to spatial-temporal variations in the thermal gradient (G) and solidification rate (R) as presented in G-R solidification maps³¹. These terms strongly depend on local composition, which is governed by liquid/solid interfacial instabilities (e.g. constitutional undercooling, CU)^{17,31}. For the hierarchical microstructure shown in Fig. 2d, CU may be able to provide the necessary driving force for grain nucleation leading to regions with fine equiaxed grains. It seems plausible to suggest that CU variations can take place locally in heat-affected regions promoting the formation of fine equiaxed α grains: cyclic re-melting of previously deposited material may improve the

compositional homogeneity of the Ti-2La powder blend, and create new nuclei via CU. The low solubility of La in α -Ti and β -Ti may as well contribute to a large CU²⁸. Although columnar grain growth is observed in Fig. 2d, zooming in a representative region of Fig. 2d (see Fig. 2e) reveals that fine equiaxed α grains (<10 μ m) coexist with the elongated and tortuous α grains, frequently decorating their boundaries. Thus, nucleation ahead of the solidification front, i.e. local alteration of the solidification mode, may be induced in these regions (Fig. 2e). Furthermore, the alternating microstructure of Fig. 2d is probably also influenced by the chequerboard SLM strategy since the scanning direction of the laser, i.e. the thermal gradient front of the melt pool, changes for each layer (Methods). Changes in the crystallographic orientation of grains have been reported for AM powder-bed techniques by changing processing parameters³². These effects do not occur for CP Ti.”

C7. * *Line 126: The authors use a rather ambiguous term "rapid". Can this be quantified with respect to time?*

In the resubmitted version of the manuscript, the authors have included the exact value following the reviewer's request:

Page 9, lines 195–197:

“As the temperature decreases, a rapid transformation $\beta \rightarrow \alpha$ leading to ~95vol.% of α in ~ 3.7 min takes place between 900–850°C.”

This calculation is derived from the evolution of the volume fraction of α as a function of a time presented in Supplementary Fig.4.

C8. * *Line 137: 5100°C/min - I believe there is a typo here.*

In order to avoid possible misleading from the previous description, the quenching conditions pointed by the reviewer have been clarified in the new version of the manuscript:

Page 10, lines 206-209:

“Metallographic analysis of the SLM Ti-2La alloy quenched from 950°C ($L_1+\beta$ field) down to RT (Methods), with a cooling rate of ~ 85°C/s between 950–350°C, is shown in Fig. 4d.”

C9. * *Line 153: Suggest replacing "irrelevant" with "negligible"*

The modification has been included in the resubmitted manuscript (Page 11, line 225).

C10. * *Lines 157-158: The authors use pole figures and multiple of random distribution (MRD) values to underpin their argument that the Burgers OR is not adopted. There is insufficient evidence that this is true. E.g., Fig. 2 (a,b) show a mrd of 2.4 and 1.9, with hot spots in similar locations. Arguably, this may be a 30-35% reduction in texture, but clearly the Burgers OR is still active. Traditional evidence would include TEM microscopy (or higher resolution EBSD), and ultimately, the authors would be more able to convincingly prove that this is true.*

The lines referred here are also pointed by the reviewer in her/his comment C5 (Reviewer#1), where the incorporation of a new figure (Supplementary Fig. 3) providing traditional evidence of possible

non-occurrence of the Burgers OR in the Ti-2La system has been presented. The reviewer is cordially referred to this modification as well as to the other new figure added “Supplementary Fig. 5”, which also provides further evidence that a non-Burgers OR can take place in the Ti-1.4Fe-1La system (comment C2 of Reviewer#1). The authors agree with the statement of the reviewer pointing that the Burgers OR is still partially active, and therefore, two distinct paths of α formation are observed: α formed via the Burgers OR and non-Burgers OR α derived from the path $L_1+\beta \rightarrow \alpha$. Thus, the text has been modified accordingly for both Ti-La and Ti-1.4Fe-1La systems:

Page 11, lines 227–234:

“The pole figures in Fig. 4e for $\{002\}\alpha$ and $\{110\}\beta$ at 850°C, i.e. at the end of the $\beta \rightarrow \alpha$ transformation, show that α forms to a large extent with orientations distinct from those given by the strong poles shown by β . This indicates that a fraction of α does not inherit the texture of β within this temperature range. This, together with local crystallographic orientation analysis of adjacent α - β grains in the condition obtained upon cooling down to RT (Supplementary Fig.3), indicates that the usual Burgers OR relationship $\{002\}\alpha \parallel \{110\}\beta$ typical for $\beta \rightarrow \alpha$ transformation is partially avoided.”

Page 14, lines 304–310:

“Accordingly, the bi-modal distribution of lattice correlation boundaries between α and β phases obtained by EBSD for this condition (Fig. 5h), shows the presence of two distinct paths of α formation: α formed via the Burgers OR and non-Burgers OR α derived from the path $L_1+\beta \rightarrow \alpha$. Moreover, Supplementary Fig. 5 provides further crystallographic evidence of non-operating Burgers OR in the heat treated Ti-1.4Fe-1La alloy by considering adjacent α - β regions.”

C11. * Lines 174 and figures: Comments on tortuous primary alpha grains. This is an interesting result. The authors are encouraged to vary the input energy to eliminate low input energies which, when coupled with elemental blends, cause challenges and local "cold spots" in the liquid as the elements separately melt and then have the thermodynamics such as the enthalpy of mixing come into play.

The authors appreciate this constructive suggestion. The SLM scanning strategy used in our investigations –as pointed in the section Methods (page 17, lines 366–372 of the resubmitted version)– aims at applying a severe intrinsic heat treatment in order to promote intensive re-melting of solute La and CP Ti powders, i.e. increase compositional homogeneity during SLM manufacturing. This methodology was chosen after obtaining formation of heterogeneities in the bulk material when parameters commonly used for synthesizing ingot-derived powder alloys were applied.

An illustrative example of this is presented in the figure below, where irregular distributions of solute La (white regions) were obtained by the authors using a laser power and hatch distance of 200W and 100 μ m, respectively, commonly used during SLM of Ti-6Al-4V⁶. Thus, aiming at avoiding “cold spots” and promoting longer interaction times between the laser and the material, a higher laser power of 350W and a tighter hatch distance of 40 μ m were applied. As shown in Fig. 1b and Fig. 2d of the manuscript, this leads to minimization of solute heterogeneities in the alloy.

The aspect pointed by the reviewer discussed in the previous lines, has been clarified in the resubmitted manuscript:

Page 17, lines 370–374:

“This strategy aims at applying a severe intrinsic heat treatment induced by the scanning laser⁶ in order to promote intensive re-melting of solute and CP Ti powders, i.e. increase the compositional homogeneity of the manufactured as-built bulk alloys. The chosen SLM parameters promoted minimization of irregular distributions of solute La.”

C12. * Line 200: The authors use the short-hand "CU" for constitutional undercooling, yet this the only time this abbreviation appears. The authors should simply use "constitutional undercooling", as there is no need for an abbreviation for one use.

Owing to the modifications introduced in the resubmitted version of the manuscript, pointed in comment C6 of Reviewer #1, the terminology “constitutional undercooling” is now used several times in the text. Thus, its abbreviation “CU” has been incorporated in the resubmitted text for referring this term after its first use in page 8 line 155.

C13. * Line 243: The authors speculate that Mg or Ca could be used, but each of these present challenges given their melting point.

Indeed, the difference between the melting point of Ti (1668°C) with respect to Mg (648°C) and Ca (842°C) is remarkable. Thus, as the reviewer correctly points, it may represent a challenge during SLM manufacturing. Since the role of these elements is not studied in our investigations, we have avoided this

speculation in the resubmitted version of the manuscript in accordance with the reviewer's suggestion. The authors cordially refer her/him to page 15, lines 324-326 of the new manuscript version with highlighted changes.

Moreover, the significance of our investigations has been better contextualized in the resubmitted text following the suggestion introduced by Reviewer #2 in comment C1:

"The general idea of adapting Ti-alloys to AM by using a peritectic reaction is absolutely novel and will open up windows for target oriented alloy design in other alloy systems as well."

The following paragraph has been modified in accordance with the inputs of Reviewer #1 and Reviewer #2:

Page 15, lines 323–327:

"The approach of adding peritectic forming elements capable to induce as-built as well as post-processed additive-layer-manufactured microstructures of reduced texture, can have a positive impact in commercial titanium compositions. Moreover, the general idea of adapting alloys to AM using a peritectic reaction can open up windows for target oriented alloy design in other alloy systems."

C14. * *Line 249: What does "argon 5.0 atmosphere employing" mean? Does this mean it was a positive pressure of 5x atmosphere?*

As clarified in the resubmitted version of the manuscript, the purity of the argon gas employed during SLM is meant here:

Page 15, lines 332–335:

"Selective laser melting (SLM) of powders of commercially pure titanium grade 1 (CP Ti, with max. 0.18wt.% O and 0.2wt.% Fe) as well as of powder blends of CP Ti-2wt.% La and CP Ti-1.4Fe-1La (wt.%), was carried out with flowing argon 5.0 (i.e. Ar purity \geq 99.999 %) at a consumption rate of \sim 2.5L/min."

2. Reviewer #2 (Remarks to the Author):

C1. *The authors report on a new Ti-base alloy for additive manufacturing (AM, also referred to as 3D printing). Even if AM already is used in industry for realization for complex components, numerous upon questions/issues remain.*

One major roadblock to more widespread applications is the limited number of alloys being available to be processed employing powder bed techniques such as selective laser melting (SLM). Most importantly, alloys employed so far have been developed for conventional processing routes, e.g. forging and casting. Thus, these alloys are not adapted to the unique processing conditions prevailing in AM and SLM, respectively. Rapid solidification and epitaxial growth are two aspects to be mentioned in this regard. In the field of Ti-base alloys, almost exclusively Ti-6Al-4V and commercial purity (CP) titanium are considered in both academia and industry. It is well accepted that these alloys suffer from several process-induced issues: anisotropic microstructure and mechanical properties as well as low damage tolerance. Thus, post treatments always need to be conducted. Still, the anisotropic nature of deformation prevails upon standard post processing routes.

In consequence, development of new Ti-alloys meeting the process conditions of AM and SLM, respectively, and allowing for realization of isotropic microstructures are crucially needed for further development in the field. By adding La to CP titanium the authors address this topic in an excellent way. Results obtained are of highest quality and were elaborated using absolutely sophisticated characterization techniques, e.g. high-energy X-ray diffraction (HEXRD). Conclusions drawn based on data presented are fully convincing.

The general idea of adapting Ti-alloys to AM by using a peritectic reaction is absolutely novel and will open up windows for target oriented alloy design in other alloy systems as well.

The thorough experimental effort, quality of data, in-depth discussion and expected impact of the approach presented are clearly up to the standard of Nature Communications.

Quality of figures and text and, thus, presentation of results is excellent.

In consequence, the reviewer strongly recommends acceptance of the current work.

In order to further strengthen their contribution, the authors should consider the following (not mandatory):

In the introduction section, numbers provided should be substantiated. Production savings up to 50% are highlighted. Which kind of components are referred to here?

The reviewer refers to lines 37–39 of the previous version the manuscript. There, it was vaguely mentioned that the advantages of AM titanium-based components account for “*estimated production savings up to 50%, by missing out in major part, exorbitant machining costs and material loss*”³. Aiming at providing the clarification required, the resubmitted version incorporates a concise paragraph including specific examples where AM leads to manufacturing savings with respect to conventional processing of Ti-alloys. For that, a new reference has been added (4. Dehoff, R. et al. Case study: additive manufacturing of aerospace brackets. *Adv. Mater. Process.* **171**, 19–22 (2013)). These modifications have been incorporated in the text as follows:

Page 3, lines 37–50:

“For titanium-based components, these advantages account for estimated production savings up to 50%, by missing out in major part, exorbitant machining costs and material loss³. In aerospace, AM of titanium components focuses on parts with high buy-to-fly ratio: the weight of purchased stock material with respect to that of the finished part. Typical aerospace components can have 10:1, 20:1, and even 40:1 buy-to-fly ratios using conventional manufacturing processes. AM is capable to reduce it to close to 1:1. For instance, 50% reduction of the production costs has been reported for a conventionally fabricated wrought Ti-6Al-4V engine bracket using AM⁴. AM also allows repair of expensive titanium-based components (e.g. flanges, fan blades, casings, vanes and landing gears) at 20-40% of the cost of the new parts¹. Beyond the manufacturing chain, AM weight-optimized components can imply a progress of environmental targets. Previous studies concluded that savings of 3.3 million litres of fuel over the aircraft’s life can be obtained by a weight reduction of 55% using AM Ti-6Al-4V seat buckles¹.”

C2. Furthermore, additional references to current literature could be provided. Realization of isotropic microstructures in Al-alloys as well as steels has been published quite recently: use of nano-sized particles for grain refinement in high-strength aluminium alloys and employment of multiple phase transformations induced by intrinsic heat treatment in high alloyed steels. These concepts should be introduced shortly in the introduction section even if not being applicable to Ti-base alloys so far.

We have added new citations reporting advances in tackling epitaxial growth of AM alloys aiming at better contextualization. This includes slight modification of the chemical composition of Ti-alloys. The reviewer is cordially referred to the comment C1 of Reviewer #1 where incorporation of this research in the manuscript is presented. Since this is not the only strategy, and as the reviewer correctly points, this subject of study requires a broader perspective of analysis, new insights using the intrinsic heat treatment as well as nanoparticles in different materials (e.g. steel, aluminium) have also been introduced in the text. Thus, new references have been incorporated:

25. Xu, W. et al. Additive manufacturing of strong and ductile Ti-6Al-4V by selective laser melting via in situ martensite decomposition. *Acta Mater.* **85**, 74–84 (2015).

26. Jäggle, E.A. et al. Precipitation reactions in age-hardenable alloys during laser additive manufacturing. *JOM* **43**, 943–949 (2016).

27. Kürsteiner, P. et al. Massive nanoprecipitation in an Fe-19Ni-xAl maraging steel triggered by the intrinsic heat treatment during laser metal deposition. *Acta Mater.* **129**, 52–60 (2017).

A paragraph discussing the new content explained in the previous lines has been included in the resubmitted manuscript:

Page 5, lines 97-110:

“A further strategy addressing the reduction of anisotropy and improvement of the strength-ductility trade off during metal-based AM consists in exploring the possibilities of the intrinsic heat treatment (IHT), namely the thermal history induced by the heating source (e.g. laser) to previously deposited layers. As shown for Ti-6Al-4V, intensified IHT can generate extensive martensite decomposition leading to configurations of stable α and β phases in a single AM process step^{6,25}. For precipitation hardened alloys, the IHT can provoke the formation of finely dispersed second phase particles and therefore, offers the possibility to shorten or avoid subsequent aging treatments²⁶. Illustrative examples of this effect are the homogeneous dispersions of fine N_3Ti and $NiAl$ precipitates obtained in maraging steels, as well as of $Al_3(Sc, Zr)$ particles in an Al-Sc alloy^{26,27}. Besides dispersion hardening effects, the incorporation of nanoparticles permitted to avoid large columnar grains and cracking during SLM of high-strength aluminium alloys. These nucleants promoted formation of fine-grained microstructures resulting in strengths comparable to that of wrought materials².”

C3. The authors should plot the binary phase diagram in a way that allows highlighting eutectic, peritectoid and eutectoid reaction in the system.

A figure magnifying the studied region of the Ti-La phase diagram has been included as new “Supplementary Fig.1” in the resubmitted manuscript. In agreement with the reviewer’s suggestion, clearer observation of the invariant reactions occurring in the studied zone is provided: namely peritectic $L_1 + \beta \leftrightarrow \text{La-bcc}$, peritectoid $\text{La-bcc} + \beta \leftrightarrow \alpha$, eutectoid $\text{La-bcc} \leftrightarrow \text{La-fcc} + \alpha$, and allotropic $\text{La-fcc} \leftrightarrow \text{La-dhcp}$, according to the current knowledge of the Ti-La equilibrium phase diagram. No eutectic reaction takes place. These equilibrium invariant reactions are pointed and contrasted in Table 1 with the transformations identified in situ using high energy synchrotron X-ray diffraction. It must be emphasized that investigation of the Ti-La phase diagram is not the focus of our work, and the data presented in the manuscript is based on other works^{28,29}, not on own investigations.

Supplementary Fig. 1 a Portion the Ti-La phase diagram and b magnified detail of the studied region adapted from^{28,29}, indicating the compositions used for selective laser melting.

Incorporation of the new Supplementary Fig. 1 has been pointed in the resubmitted text:

Page 6, lines 122-126:

“According to the current knowledge of the Ti-La equilibrium phase diagram (shown partially in Fig. 1a and Supplementary Fig. 1), at 2wt.% La the Ti-La system presents during cooling two paths of α formation after passing through a $L_1 + \beta \rightarrow \text{La-bcc}$ peritectic reaction²⁸: $\text{La-bcc} + \beta \rightarrow \alpha$ (peritectoid) and $\text{La-bcc} \rightarrow \text{La-fcc} + \alpha$ (eutectoid).”

C4. *Cost for the element La should be provided.*

The cost of La powder used in our investigations has been included in the resubmitted version of the manuscript. It is worth noting that the price strongly varies depending on the element form (e.g. powder or pieces). This is reflected in the diagram provided below obtained from the CES Edupack 2017 software, where the price of La is presented with respect to commonly used alloying elements in titanium. For instance, La can be significantly cheaper than V, alloying element of the most commercialized Ti alloy Ti-6Al-4V. This reference has been included in the resubmitted manuscript. Thus, reduction of the cost impact of La via ingot metallurgy can be expected, i.e. the use of ingot-based powder alloys instead of powder blending for additive manufacturing may be more economically viable.

The explanation of the previous lines has been reflected in the resubmitted new version of the text:

Page 16, lines 353–358:

“The SLM equipment was supplied by SLM solutions GmbH, CP Ti powder produced by gas atomization by TLS Technik GmbH, La (packaged in Ar atmosphere) and Fe powders by Alfa Aesar and BASF CEP SM, respectively. The cost of La powder was 23 €/g. However, the price of La strongly varies depending on the element form (e.g. powder compared to pieces). For instance, it can be significantly cheaper than V, alloying element of the most commercialized Ti alloy Ti-6Al-4V⁴⁵.”

C5. *Some details mentioned in the text (e.g. agglomerations of alpha' plates (page 5, line 98)) cannot directly be seen in the corresponding figures. The authors should add further markers to highlight these features.*

New indications have been added to the resubmitted Fig. 2 to better indicate the features mentioned by the reviewer. She/he is cordially referred to her/his comment C10 (Reviewer #2), where this and other changes requested for Fig. 2 can be observed. The new indication is reflected in the resubmitted text:

Page 7, lines 148–151:

“Minor agglomerations of α' plates can also be seen as pointed by arrows in the magnified region of Fig. 2d (see Fig. 2e). Arrangements of fine α grains can be observed between layers of elongated α grains marked between discontinuous lines in Fig. 2d.”

C6. *The authors should consider showing (part of) the phase diagram for the ternary Ti-Fe-La system in the supplement.*

The authors would like to point that the ternary Ti-Fe-La is an unusual system, as mentioned by Reviewer#1 in comment C1 as well. To the best of our knowledge (and after intensive literature research), there are no reports available on the Ti-Fe-La phase diagram. In the following, we include the only reported work dealing with the Ti-Fe-La system. However, it is not focused on the determination of the phase diagram itself and it does not offer any additional information in understanding the effect of La in alloys containing beta stabilizers such as Fe, which is the aim of our investigations.

Wang, X., Chen, R., Chen, C., Wang, Q. Hydrogen storage properties of $Ti_xFe + y$ wt.% La and its use in metal hydride hydrogen Compressor. *J. Alloys Compd.* **425**, 291–295 (2006).

Unfortunately, the data presented in this reference is so scarce that we have decided not to include it in the new version of the manuscript. The lack of previous investigations reporting the ternary (or part of) the Ti-Fe-La diagram, has been indicated in the resubmitted manuscript:

Page 13, lines 290–292:

“To the best of the author’s knowledge, no data about the ternary Ti-Fe-La phase diagram has been reported for the Ti-rich corner.”

C7. *When discussing results shown (or in the corresponding part of the methods section) clear statements on number of samples and sample volume probed have to be provided.*

The additional information requested by the reviewer has been included in the Methods section of the resubmitted manuscript:

Page 19, lines 417–419:

“The investigated CP Ti, Ti-2La and Ti-1.4Fe-1La alloys were probed by in situ HEXRD using a gauge volume of $1 \times 1 \times 5 \text{ mm}^3$.”

Page 18, lines 393–394:

“Distribution of lattice correlation boundaries between α and β phases was performed for an area of $350 \times 250 \mu\text{m}^2$.”

Page 18, lines 399–402:

“Line sequences of indentations covering the complete height of the sample (from $z = 0 \text{ mm}$ to $z = 10 \text{ mm}$) were carried out using a force of 200g ($HV_{0.2}$). The values obtained correspond to an average of 110 different indentations taken along z .”

Page 16, lines 362–370:

“Seven cubes of $10\times 10\times 10\text{mm}^3$ were built on support structures (*height*= 1mm) in a single SLM manufacturing process using a chequerboard scanning with an increment of 90° from layer to layer. Three SLM manufacturing jobs were carried out independently for CP Ti, Ti-2La and Ti-1.4Fe-1La. These sample dimensions permit evaluating SLM synthesized microstructures from different powder blends. The following are the main SLM processing parameters employed for the studied alloys: *laser power*= 350W, *scanning velocity*= 1000mm/s, *hatch distance*= $40\mu\text{m}$, *focal offset distance*= 0mm and *layer thickness*= $50\mu\text{m}$, resulting in a *volume energy density* = $175\text{J}/\text{mm}^3$.”

C8. *Using powder blends for AM always opens up the following questions: What are the characteristics of all powders employed (details on La and Fe powders are missing). Using which apparatus the powders were mixed? Are all samples processed homogeneous in terms of chemical composition?*

The authors would like to kindly refer the reviewer to comment C11 of Reviewer #1, where compositional homogeneity during in situ alloying of powders during SLM has been discussed by linking the SLM scanning strategy employed with a new figure provided. In addition to this, the information requested by the reviewer, namely the characteristics of powders and blending process, has been included in the resubmitted text:

Page 16, lines 344–355:

“The alloys were produced by blending base powder of CP Ti with additions of commercial powder of pure La (99.9%) and Fe (>99%) elements with maximal and mean particle sizes of $\sim 74\mu\text{m}$ and $\sim 3.5\mu\text{m}$, respectively. Powder blending was performed inside stainless steel containers within a glovebox kept in an atmosphere of argon 5.0 of purity and <1ppm of oxygen content. Thereafter, flowability tests were successfully carried out employing a stand Ti-6Al-4V funnel for testing free-flowing metal powder according to ISO 4490:2001. The containers –of $\sim 12\text{kg}$ of powder capacity– were tightly sealed by using a valve inside the glovebox. By doing so, they were prepared to be placed in the SLM machine.

The SLM equipment was supplied by SLM solutions GmbH, CP Ti powder produced by gas atomization by TLS Technik GmbH, La (packaged in Ar atmosphere) and Fe powders by Alfa Aesar and BASF CEP SM, respectively.”

C9. *Regarding SLM processing the following questions should be answered: Is any information on element evaporation available? What size of build envelope was employed? Is there any information on gas flow?*

There is no experimental data available about material evaporation during SLM. However, a critical role of this effect is not expected compared to Ti-6Al-4V, since the energy density employed ($175\text{J}/\text{mm}^3$) in our investigations is within those being used for this alloy (e.g. ^{6,11,25}), and the boiling temperature of Ti (3287°C) is closer to those of the alloying elements employed in our investigations, namely La (3464°C) and Fe (2861°C), than that of the main alloying element in Ti-6Al-4V, Al (2519°C)⁴⁴. This, and the other requested information (concerning dimensions of the build envelope and gas) have been clarified in the manuscript:

“Selective laser melting (SLM) of powders of commercially pure titanium grade 1 (CP Ti, with max. 0.18wt.% O and 0.2wt.% Fe) as well as of powder blends of CP Ti-2wt.% La and CP Ti-1.4Fe-1La (wt.%), was carried out with flowing argon 5.0 (i.e. purity >99.999 %) at a consumption rate of ~ 2.5L/min. A SLM 280HL machine incorporating an in-situ melt pool monitoring system that can detect hot/cold spots during SLM was employed. The temperature of the building platform made of Ti-6Al-4V (wt.%) and the oxygen content were < 40°C and < 0.14%, respectively. The building platform was not externally heated during SLM processing. A reduced build envelope of 50×50×50mm³ was used. The boiling point, t_b , of Ti, La and Fe is 3287, 3464 and 2861°C, respectively⁴⁴. Compared to the classical Ti-6Al-4V (with Al, t_b = 2519°C), a less critical role of element evaporation is expected for the studied CP Ti, Ti-2La and Ti-1.4Fe-1La alloys under the same processing conditions.”

C10. Scale bars in Figure 2 remain unclear: Scale bars of 100 microns and 50 microns seem to be valid for *c* and *d*, respectively. The overall size of both figures is very similar. The boxes highlighting the areas depicted in high resolution at the bottom are clearly different in size in the overview images. Thus, the same scale bar cannot be correct for *c* and *d*. Please clarify.

The authors appreciate the reviewer’s careful assessment, and would like to refer him to the new Fig. 2, where the size of the boxes indicating the magnified regions has been corrected:

C11. Finally, showing a stress-strain curve from tensile testing would be very helpful to draw a complete picture of the new material processed. Are data available?

Data on mechanical testing is unfortunately unavailable yet. We are well aware of the significance of such information. However, this would require the production of much larger samples, which is planned for a further project on these alloys and for which we are waiting for funding. We will certainly present mechanical testing data in future investigations.

3. Reviewer #3 (Remarks to the Author):

C1. This is a well-written paper with an interesting result, namely that *La* can be useful for changing the solidification (phase) path in titanium. I appreciate the use of synchrotron x-ray diffraction experiments to complement the microstructural characterization. Unfortunately, justifying publication in *Nature Communications* on the basis of mitigating strong texture in typical additively printed Ti-6Al-4V is not supported by the literature because, in fact, the texture in the dominant (~90 %) alpha phase is weak; see, e.g., A. A. Antonysamy, J. Meyer, and P. B. Prangnell, *Effect of build geometry on the beta-grain structure and texture in additive manufacture of Ti-6Al-4V by selective electron beam melting*, *Materials Characterization*, 84, 153-168 (2013). That the texture is weak is unsurprising because when the strongly columnar (bcc) beta transforms to (hexagonal) alpha following the Burgers orientation relationship, the high driving force available under rapid cooling means that there is negligible variant selection and the texture in the alpha is weak.

We have to emphasize this issue: **texture mitigation in Ti-alloys produced by additive manufacturing is a very timely and relevant matter, both from a technical and scientific point of view.** We take here the liberty of quoting some comments given by the other reviewers of our work:

*“The general concept of **breaking up texture in additively manufactured titanium alloys is of broad interest** to the community”*

*“What is most interesting is the **control of texture** through the peritectic reaction. The authors have investigated a seemingly unique space, and have **added to the overall field.**”*

*“...alloys employed so far have been developed for conventional processing routes, e.g. forging and casting. Thus, these alloys are not adapted to the unique processing conditions prevailing in AM and SLM. Rapid solidification and **epitaxial growth** are two aspects to be mentioned in this regard.*

*“...**It is well accepted** that these alloys suffer from several process-induced issues: **anisotropic microstructure** and mechanical properties as well as low damage tolerance. Thus, post treatments always need to be conducted. Still, the **anisotropic nature of deformation** prevails upon standard post processing routes.”*

Moreover, we provide a list of relevant and recent references dealing with this issue. This is by far not a complete literature review on this matter; it is only intended to reflect the relevance of texture formation and α texture in AM Ti alloys:

- Antonysamy, A.A., Meyer, J. & Prangell, P.B. Effect of build geometry on the β -grain structure and texture in additive manufacture of Ti-6Al-4V by selective electron beam melting. *Mater. Charact.* **84**, 153-168 (2013).
- Collins, P.C. et al. Microstructural control of additively manufactured metallic materials. *Annu. Rev. Mater. Res.* **46**, 1–18 (2016).
- Gorsse, S., Hutchinson, C., Gouné, M. & Banerjee, R. Additive manufacturing of metals: a brief review of the characteristic microstructures and properties of steels, Ti-6Al-4V and high-entropy alloys. *Sci. Tech. Adv. Mater.* **18**, 584–610 (2017).
- Kok, Y. et al. Anisotropy and heterogeneity of microstructure and mechanical properties in metal additive manufacturing: A critical review. *Mater. Des.* **139**, 565–586 (2018).
- Lu, J. et al. In-situ investigation of the anisotropic mechanical properties of laser direct metal deposition Ti6Al4V alloy. *Mater. Sci. Eng. A* **712**, 199–205 (2018).
- Saboori, A. et al. An overview of additive manufacturing of titanium components by directed energy deposition: microstructure and mechanical properties. *Appl. Sci.* **883**, 1–23 (2017).

Please note that we have even included the reference pointed by the reviewer in which texture in α is clearly shown. We reproduce here the pole figures of α and β presented in this work, reference⁵ of our manuscript (it must be mentioned that the pole figure of β was calculated and not measured experimentally):

The texture of α is clearly visible and corresponds to the typical texture observed owing to epitaxial growth and Burgers OR in AM Ti alloys. Moreover, the authors of this publication write in section 3.2.2:

“In general, the α -transformation textures were far weaker than their reconstructed β -parent textures and had a maximum intensity of only ~ 3 times random in bulk sections, compared to 8 times random seen for the parent β -texture.”

It is evident that the authors are comparing the pole figures of α and β , and therefore, they mention that texture of α is weaker **than that of** β , i.e. from a relative point of view to the parent phase, which is related to the nature of the β to α transformation in Ti alloys, or, as the authors put it:

*“Because of crystal symmetry, this provides 12 possible variant orientations that can form from a single parent β -grain, which if randomly selected will dilute the texture **compared to the β -solidification texture.**”*

Again, they compare the texture between α and β and, while it can be argued that the texture of α is weaker than that of β it must certainly be said that α is still strongly textured and this agrees with the results shown in the mentioned paper (and in many of the reports given in the list above).

Therefore, we disagree with the argumentation of the reviewer and kindly ask her/him to reconsider her/his position.

C2. *Also, martensitic structures can be easily reverted to two-phase via heat treatment which is generally done if for no other reason than stress relief, especially in laser powder bed materials.*

The aspect pointed by the reviewer has been discussed in the resubmitted version of the manuscript. The reviewer is cordially referred to comment C2 of Reviewer #2 and comment C3 of Reviewer #1.

C3. *A second criticism concerns the interpretation of the tortuous grain shapes that were observed: the authors need to read the literature on 3D printing and check what scan strategy was used in their particular builds because it can happen that offsets and changes in direction force the direction of the thermal gradient (and therefore the solidification path) to change substantially from layer to layer, resulting in the observed pattern. By contrast, scanning in the same direction in successive layers with no offset in lateral position can result in very strong epitaxial growth. This means that texture is at least as much influenced by scan strategy as by local solidification conditions. I apologize for not offering more detailed feedback but the authors need to re-think the basis for the originality of their work.*

The criticism introduced by the reviewer has been also discussed in the resubmitted version of the manuscript by taking into account the support of the literature and the helpful contribution of Reviewer 1. This is clarified in comment C6 of Reviewer 1, to which the reviewer is kindly referred.

At last, the authors would like to thank the reviewers for careful reading of the manuscript and the provision of their constructive comments.

REVIEWERS' COMMENTS:

Reviewer #1 (Remarks to the Author):

Thank you for the careful attention to the reviewers comments.

Reviewer #2 (Remarks to the Author):

The authors report on a new Ti-base alloy for additive manufacturing (AM, also referred to as 3D printing). Even if AM already is used in industry for realization for complex components, numerous upon questions/issues remain.

One major roadblock to more widespread applications is the limited number of alloys being available to be processed employing powder bed techniques such as selective laser melting (SLM). Most importantly, alloys employed so far have been developed for conventional processing routes, e.g. forging and casting. Thus, these alloys are not adapted to the unique processing conditions prevailing in AM and SLM, respectively. Rapid solidification and epitaxial growth are two aspects to be mentioned in this regard. In the field of Ti-base alloys, almost exclusively Ti-6Al-4V and commercial purity (CP) titanium are considered in both academia and industry. It is well accepted that these alloys suffer from several process-induced issues: anisotropic microstructure and mechanical properties as well as low damage tolerance. Thus, post treatments always need to be conducted. Still, the anisotropic nature of deformation prevails upon standard post processing routes.

In consequence, development of new Ti-alloys meeting the process conditions of AM and SLM, respectively, and allowing for realization of isotropic microstructures are crucially needed for further development in the field. By adding La to CP titanium the authors address this topic in an excellent way. Results obtained are of highest quality and were elaborated using absolutely sophisticated characterization techniques, e.g. high-energy X-ray diffraction (HEXRD). Conclusions drawn based on data presented are fully convincing.

The general idea of adapting Ti-alloys to AM by using a peritectic reaction is absolutely novel and will open up windows for target oriented alloy design in other alloy systems as well.

The thorough experimental effort, quality of data, in-depth discussion and expected impact of the approach presented are clearly up to the standard of Nature Communications.

Quality of figures and text and, thus, presentation of results is excellent.

In consequence, the reviewer strongly recommends acceptance of the current work in its revised version.

Reviewer #3 (Remarks to the Author):

I appreciate the effort that the authors have made to respond to the feedback and improve the manuscript. I have not changed my mind about whether it belongs in Nature Comm. because it does not. Instead it should go to one of the metallurgical journals. The discussion of texture and anisotropy is naive. Yes there is anisotropy but it is a second order problem. The texture of the alpha is not strong, precisely because of the martensitic phase transformation and formation of multiple variants. Heat treatments of martensitic Ti alloy can yield impressive ductility, see the work by Charlotte de Formanoir, e.g. Rejection is recommended.

1. Reviewer #1 (Remarks to the Author):

C1. *Thank you for the careful attention to the reviewers comments.*

2. Reviewer #2 (Remarks to the Author):

C1. *The authors report on a new Ti-base alloy for additive manufacturing (AM, also referred to as 3D printing). Even if AM already is used in industry for realization for complex components, numerous upon questions/issues remain.*

One major roadblock to more widespread applications is the limited number of alloys being available to be processed employing powder bed techniques such as selective laser melting (SLM). Most importantly, alloys employed so far have been developed for conventional processing routes, e.g. forging and casting. Thus, these alloys are not adapted to the unique processing conditions prevailing in AM and SLM, respectively. Rapid solidification and epitaxial growth are two aspects to be mentioned in this regard. In the field of Ti-base alloys, almost exclusively Ti-6Al-4V and commercial purity (CP) titanium are considered in both academia and industry. It is well accepted that these alloys suffer from several process-induced issues: anisotropic microstructure and mechanical properties as well as low damage tolerance. Thus, post treatments always need to be conducted. Still, the anisotropic nature of deformation

prevails upon standard post processing routes.

In consequence, development of new Ti-alloys meeting the process conditions of AM and SLM, respectively, and allowing for realization of isotropic microstructures are crucially needed for further development in the field. By adding La to CP titanium the authors address this topic in an excellent way. Results obtained are of highest quality and were elaborated using absolutely sophisticated characterization techniques, e.g. high-energy X-ray diffraction (HEXRD). Conclusions drawn based on data presented are fully convincing.

The general idea of adapting Ti-alloys to AM by using a peritectic reaction is absolutely novel and will open up windows for target oriented alloy design in other alloy systems as well.

The thorough experimental effort, quality of data, in-depth discussion and expected impact of the approach presented are clearly up to the standard of Nature Communications.

Quality of figures and text and, thus, presentation of results is excellent.

In consequence, the reviewer strongly recommends acceptance of the current work in its revised version.

3. Reviewer #3 (Remarks to the Author):

C1. *I appreciate the effort that the authors have made to respond to the feedback and improve the manuscript. I have not changed my mind about whether it belongs in Nature Comm. because it does not. Instead it should go to one of the metallurgical journals. The discussion of texture and anisotropy is naive. Yes there is anisotropy but it is a second order problem. The texture of the alpha is not strong, precisely because of the martensitic phase transformation and formation of multiple variants.*

We have to emphasize this issue: **texture mitigation in Ti-alloys produced by additive manufacturing is a very timely and relevant matter, both from a technical and scientific point of view.** In the last revision, the reviewer was cordially referred to the comments given by Reviewer #1 and Reviewer #2 explaining this problematic, as well as to a list of illustrative references –besides those included in the introduction– reflecting the relevance of texture formation in AM Ti alloys. The one reference pointed by the reviewer to argue weak texture of α in AM Ti alloys was also included in this list, since this texture was clearly visible in the pole figures of the pointed work, provided in the previous revision. This is a consequence of epitaxial growth and Burgers OR in AM Ti alloys. As discussed in our previous answer to the reviewer, she/he is comparing the texture between α and β and, while it can be argued that the texture of α is weaker than that of β it must certainly be said that α is still strongly textured. This agrees with the results shown in the mentioned references.

C2. *Heat treatments of martensitic Ti alloy can yield impressive ductility, see the work by Charlotte de Formanoir, e.g. Rejection is recommended.*

The reviewer refers to the following work:

- De Formanoir, C., Michotte, S., Rigo, O., Germain, L. & Godet, S. Electron beam melted Ti-6Al-4V: microstructure, texture and mechanical behavior of the as-built and heat-treated material. *Mater. Sci. Eng. A* **652**, 105–119 (2016).

The pointed investigations reflect again the general problematic of tackling epitaxial growth during AM of Ti-alloys, as discussed in the previous comment C1 of the reviewer and in the previous revision. Here, we reproduce the pole figures of α and β presented in this work reflecting this effect (it must be mentioned that the pole figure of β was calculated and not measured experimentally):

The texture of α and β is clearly visible and corresponds to the typical texture observed owing to epitaxial growth and Burgers OR in AM Ti-alloys. In this work, the authors apply subtransus and supertransus post-thermal treatments –i.e. below and above the β -transus of Ti-6Al-4V– to modify this microstructure obtained upon electron beam melting. As pointed in our manuscript (Page 4, lines 75–85), and was discussed in the previous revision (comment C3 of Reviewer #1), these strategies correspond to commonly applied treatments to improve the strength-ductility trade-off of as-built AM components, that do not mitigate crystallographic texture and can lead to excessive coarsening, or as the authors of the referred publication put it:

“...subtransus heat treatments only induce very moderate microstructural changes, resulting in a limited mechanical effect. Supertransus heat treatments, on the other hand, generate substantial microstructural changes. The columnar morphology is transformed into an equiaxed one, resulting in a more isotropic material. However, the fast β grain growth that these treatments induce is uncontrollable and mechanically undesirable.”

“Subtransus heat treatments have a limited impact on the microstructure of the material. The columnar morphology is maintained, and the width of the β columns observed in untreated EBM parts remains unchanged.”

“...in the case of supertransus treatments, rapid growth of the β grains occurs, leading to the formation of equiaxed β grains, which can be up to 1 mm large.”

In the resubmitted manuscript, these aspects have been better clarified and the reference pointed by the reviewer has been included¹⁵:

Page 4, lines 75–85:

“Post-thermal and/or thermomechanical treatments are commonly applied to as-built AM components to improve the strength-ductility trade-off. This can include supertransus or subtransus heat treatments^{14,15} as well as hot isostatic pressing¹⁶ inducing formation of stable α and β via decomposition of metastable microstructures. Subtransus treatments have limited impact on the microstructure and columnar morphologies derived from epitaxial growth are usually maintained. During supertransus treatments, rapid growth of β takes place leading to excessive grain growth and coarsening¹⁵. Apart from representing a costly methodology that reduces the economical attractiveness of AM, these post-treatments do not represent an alternative to mitigate crystallographic texture and its effect on mechanical performance of the alloys¹⁷⁻¹⁹.”

Therefore, we disagree again with the argumentation of the reviewer, insisting in that texture mitigation in Ti-alloys produced by AM is a very timely and relevant matter.

At last, the authors would like to thank the reviewers for careful revision of the manuscript and the provision of their constructive comments.